# CAVIA: CAMERA-CONTROLLABLE MULTI-VIEW VIDEO DIFFUSION WITH VIEW-INTEGRATED ATTENTION

## ABSTRACT

In recent years there have been remarkable breakthroughs in image-to-video generation. However, the 3D consistency and camera controllability of generated frames have remained unsolved. Recent studies have attempted to incorporate camera control into the generation process, but their results are often limited to simple trajectories or lack the ability to generate consistent videos from multiple distinct camera paths for the same scene. To address these limitations, we introduce **Cavia**, a novel framework for camera-controllable, multi-view video generation, capable of converting an input image into multiple spatiotemporally consistent videos. Our framework extends the spatial and temporal attention modules into view-integrated attention modules, improving both viewpoint and temporal consistency. This flexible design allows for joint training with diverse curated data sources, including scene-level static videos, object-level synthetic multi-view dynamic videos, and real-world monocular dynamic videos. To our best knowledge, Cavia is the first of its kind that allows the user to precisely specify camera motion while obtaining object motion. To the best of our knowledge, Cavia is the first framework that enables users to generate multiple videos of the same scene with precise control over camera motion, while simultaneously preserving object motion. Extensive experiments demonstrate that Cavia surpasses state-of-the-art methods in terms of geometric consistency and perceptual quality.

## 1 INTRODUCTION

The rapid development of diffusion models has enabled significant advancements in video generative models. Early efforts have explored various approaches, either training a video model from scratch or by fine-tuning pre-trained image generation models with additional temporal layers (Stability, 2023; Wang et al., 2023a; Ho et al., 2022b; Singer et al., 2022; Ho et al., 2022a; Nan et al., 2024). The training data of these video models typically consist of a curated mixture of image (Schuhmann et al., 2022) and video datasets (Bain et al., 2021; Wang et al., 2023b;a; Nan et al., 2024). While substantial progress has been made in improving model architectures and refining training data, relatively little research has been conducted on the 3D consistency and camera controllability of generated videos.

To tackle this issue, several recent works (Wang et al., 2023c; He et al., 2024; Bahmani et al., 2024; Xu et al., 2024; Hou et al., 2024) have attempted to introduce camera controllability in video generation, aiming to ensure that generated frames adhere to viewpoint instructions, thereby improving 3D consistency. These works either enhance viewpoint control through better conditioning signals (Wang et al., 2023c; He et al., 2024; Bahmani et al., 2024) or by utilizing geometric priors, such as epipolar constraints (Xu et al., 2024) or explicit 3D representations (Hou et al., 2024). However, despite these efforts, the generated videos often lack precise 3D consistency or are restricted to static scenes with little to no object motion. Moreover, it remains challenging for monocular video generators to produce multi-view consistent videos of the same scene from different camera trajectories.

Since independently sampling multiple sequences often results in significantly inconsistent scenes, generating multiple video sequences simultaneously is desirable. However, this remains extremely challenging due to the scarcity of multi-view video data in the wild, leading to multi-view generations limited to inconsistent near-static scenes or synthetic objects. A concurrent work, CVD (Kuang et al., 2024), builds on multi-view static videos (Zhou et al., 2018) and warping-augmented monocular videos (Bain et al., 2021), but it can only generate videos with limited baselines, yielding inconsistent

results when object motion is present. Another concurrent work, Vivid-ZOO (Li et al., 2024a), leverages dynamic objects from Objaverse (Deitke et al., 2023b) dataset and renders multi-view videos to train a video generator. However, due to limited data sources, their results are primarily object-centric frames from fixed viewpoints, lacking realistic backgrounds.

To address these challenges, we propose **Cavia**, a novel framework that extends a monocular video generator (Stability, 2023) to generate multi-view consistent videos with precise camera control. We enhance the spatial and temporal attention modules to cross-view and cross-frame 3D attentions respectively, improving consistency across both viewpoints and frames. Our model architecture enables a novel joint training strategy that fully utilizes static, monocular, and multi-view dynamic videos. Static videos (Zhou et al., 2018; Yu et al., 2023; Xia et al., 2024; Reizenstein et al., 2021; Deitke et al., 2023b;a) are converted to multi-view formats to ensure the geometric consistency in the generated frames. We then incorporate rendered synthetic multi-view videos of dynamic 3D objects (Liang et al., 2024; Jiang et al., 2024; Li et al., 2024c) to teach the model to generate reasonable object motion. To prevent overfitting on synthetic data, we finetune the model on pose-annotated monocular videos (Wang et al., 2023b; Nan et al., 2024) to enhance performance on complex scenes. Our framework synthesizes cross-view and cross-frame consistent videos, and extensive evaluations on real and text-to-image generated images show its applicability across challenging indoor, outdoor, object-centric, and large-scene cases. We systematically measure the quality of the generated videos in terms of per-video and cross-view geometric consistency and perceptual quality. Our experiments demonstrate our superiority compared to previous works both qualitatively and quantitatively. Our experiments demonstrate superior performance compared to previous methods, both qualitatively and quantitatively. Additionally, we show that our method can extrapolate to generate four views during inference and enable 3D reconstruction of the generated frames.

Our main contributions can be summarized as follows,

- We propose a novel framework, **Cavia**, for generating multi-view videos with camera controllability. We introduce view-integrated attentions, namely cross-view and cross-frame 3D attentions, to enhance consistency across viewpoints and frames.
- We introduce an effective joint training strategy that leverages a curated mixture of static, monocular dynamic, and multi-view dynamic videos, ensuring geometric consistency, high-quality object motion, and background preservation in the generated results.
- Our experiments demonstrate superior geometric and perceptual quality in both monocular video generation and cross-video consistency compared to baseline methods. Additionally, our flexible framework can operate on four views at inference, offering improved view consistency and enabling 3D reconstruction of the generated frames.

## 2 RELATED WORKS

### 2.1 CAMERA CONTROLLABLE VIDEO DIFFUSION MODELS

Recent advancements in video diffusion models have significantly benefited from scaling model architectures and leveraging extensive datasets (Bain et al., 2021; Wang et al., 2023b;a), leading to impressive capabilities in generating high-quality videos (Stability, 2023; Ho et al., 2022b; Singer et al., 2022; Ho et al., 2022a; OpenAI). While large foundational video diffusion models exist, our work focuses on enhancing camera control over video diffusion processes, a rapidly growing area of research. AnimateDiff (Guo et al., 2023) and Stable Video Diffusion (SVD) (Stability, 2023) employ individual camera LoRA (Hu et al., 2021) models for specific camera motions. MotionCtrl (Wang et al., 2023c) improves flexibility by introducing camera matrices, while CameraCtrl (He et al., 2024), CamCo (Xu et al., 2024), and VD3D (Bahmani et al., 2024) enhance the camera control accuracy by introducing Plücker coordinates to the video models via controlnet (Zhang & Agrawala, 2023). To further improve the geometric consistency, CamCo (Xu et al., 2024) applies epipolar constraints and CamTrol (Hou et al., 2024) incorporates 3D Gaussians (Kerbl et al., 2023). However, these methods focus on monocular video generation, limiting their ability to sample multiple consistent video sequences of the same scene from distinct camera paths. CVD (Kuang et al., 2024) extends CameraCtrl (He et al., 2024) for multi-view video generation, but their results are constrained to simple camera and object motion. ViVid-Zoo (Li et al., 2024a) extends MVDream (Shi et al., 2023b) for multi-view purposes but is limited to object-centric results with fixed viewpoints. In contrast, our work explores

view-integrated attentions for more precise camera control over arbitrary viewpoints and introduces a joint training strategy leveraging data mixtures to improve novel-view performance in complex scenes.

## 2.2 Multi-view Image Generation

Early approaches such as MVDiffusion (Tang et al., 2023) focused on generating multiview images in parallel by employing correspondence-aware attention mechanisms, enabling effective cross-view information interaction, particularly for textured scene meshes. Recent approaches like Zero123++ (Shi et al., 2023a), Direct2.5 (Lu et al., 2024), Instant3D (Li et al., 2023), MVDream (Shi et al., 2023b), MVDiffusion++ (Tang et al., 2024), CAT3D (Gao et al., 2024), and Wonder3D (Long et al., 2024) have introduced single-pass frameworks for multiview generation, utilizing multiview self-attention to improve viewpoint consistency. Other works, such as SyncDreamer (Liu et al., 2023b), One-2-3-45 (Liu et al., 2024), Cascade-Zero123 (Chen et al., 2023) and ConsistNet (Yang et al., 2024a), incorporate multiview features into 3D volumes to facilitate 3D-aware diffusion models (Liu et al., 2023a; Watson et al., 2022). Meanwhile, techniques such as Pose-Guided Diffusion (Tseng et al., 2023), Era3D (Li et al., 2024b), Epidiff (Huang et al., 2024), and SPAD (Kant et al., 2024) have integrated epipolar-based features to facilitate enhanced viewpoint fusion within diffusion models. Finally, approaches like V3D (Chen et al., 2024b), IM-3D (Melas-Kyriazi et al., 2024), SV3D (Voleti et al., 2024) and Vivid-1-to-3 (Kwak et al., 2024) leverage priors from video diffusion models to achieve multiview generation with improved consistency. However, these methods focus on generating static 3D objects or scenes, while our work introduces vivid object motion into multiview dynamic video generation in complex scenes.

## 2.3 4D Generation

Recent efforts in 4D generation have explored various methods (Singer et al., 2023; Zhao et al., 2023; Bahmani et al., 2023; Zheng et al., 2023; Ling et al., 2023a) that use score distillation from video diffusion models to optimize dynamic NeRFs or 3D Gaussians for text- or image-conditioned scenes. Follow-up works (Jiang et al., 2023; Ren et al., 2023; Yin et al., 2023; Ren et al., 2024; Zeng et al., 2024; Pan et al., 2024) investigate video-to-4D generation, enabling controllable 4D scene generation from monocular videos. More recent methods (Liang et al., 2024; Xie et al., 2024; Zhang et al., 2024) utilize video diffusion models to address the spatial-temporal consistency required for efficient 4D generation. However, these approaches primarily focus on object-centric generation and face challenges in producing realistic results with complex backgrounds. In contrast, our work emphasizes generating multi-view, 3D-consistent videos for complex scenes.

## 3 Method

### 3.1 Overview

Image-to-video generation takes a single image $I_0$ as input and outputs a video sequence $O_1, \cdots, O_n$. By introducing camera control, the model additionally takes in a sequence of camera information $C_1, \cdots, C_n$, which dictates the desired viewpoint changes for the output sequence. In the multi-view scenario, we extend each batch of the camera control signal and output video sequence to $V$ sequences. In the following paragraphs, we present our proposed **Cavia** framework in detail. First, we outline the preliminaries of image-to-video diffusion and describe how camera controllability is introduced in monocular video generation. Then, we elaborate on the model design for multi-view consistent video generation. An overview of our framework is provided in Fig. 1.

### 3.2 Camera Controllable Video Diffusion Model

**Preliminaries** Our model builds on pre-trained Stable Video Diffusion (SVD) (Stability, 2023). SVD extends Stable Diffusion 2.1(Rombach et al., 2022) by adding temporal convolution and attention layers, following the VideoLDM architecture (Blattmann et al., 2023). SVD is trained with a continuous-time noise scheduler (Karras et al., 2022). In each iteration, the training data is perturbed by Gaussian noise $\mathbf{n}(t) \sim \mathcal{N}(0, \sigma^2(t)\mathbf{I})$ and the diffusion model is tasked with estimating the clean data $x_0 \sim p_0$. Let $p(\mathbf{x}; \sigma(t))$ denote the marginal probability of noisy data $\mathbf{x}_t = \mathbf{x}_0 + \mathbf{n}(t)$, the iterative refinement process of diffusion model corresponds to the probability flow ordinary

Figure 1: An overview of Cavia is shown in (a). We introduce view-integrated attention modules, namely cross-view attentions and cross-frame attentions, which enforce viewpoint and temporal consistency of the generated frames, respectively. As illustrated in (b) and (c), our view-integrated attention incorporates additional feature dimensions into the attention mechanism, enhancing consistency across views and frames.

differential equation (ODE):

$$dx = -\dot{\sigma}(t)\sigma(t)\nabla_x \log p(x; \sigma(t))dt. \tag{1}$$

$\nabla_x \log p(x; \sigma(t))$ refers to the score function, which is parameterized by a denoiser $D_\theta$ through $\nabla_x \log p(x; \sigma) \approx (D_\theta(x; \sigma) - x)/\sigma^2$. We follow the EDM-preconditioning framework (Karras et al., 2022; Stability, 2023) and parameterize $D_\theta$ with a neural network $F_\theta$ as follows,

$$D_\theta = c_{\text{skip}}x + c_{\text{out}}F_\theta(c_{\text{in}}x; c_{\text{noise}}). \tag{2}$$

During training, the network $F_\theta$ is optimized using denoising score matching for $D_\theta$:

$$\mathbb{E}\left[\|D_\theta(x_0 + n; \sigma, \text{cond}) - x_0\|_2^2\right]. \tag{3}$$

**Camera Conditioning**    Although SVD is pre-trained on various high-quality video and image data, it does not natively support precise camera control instructions directly. To address this, we introduce camera conditioning to the model via Plücker coordinates (Jia, 2020), which is widely adopted as position embeddings in 360° unbounded light fields(Sitzmann et al., 2021). Plücker coordinates are defined as $P = (d', o \times d')$, where $\times$ is the cross product and $d'$ refers to the normalized ray direction $d' = \frac{d}{||d||}$. Let camera extrinsic matrix be $E = [\mathbf{R}|\mathbf{T}]$ and intrinsic matrix be $\mathbf{K}$, the ray direction $d_{x,y}$ for 2D pixel located at $(x, y)$ is formulated as $d = \mathbf{R}\mathbf{K}^{-1}\begin{pmatrix} x \\ y \\ 1 \end{pmatrix} + \mathbf{T}$. These spatial Plücker coordinates are concatenated channel-wise with the original latent inputs of SVD. We enlarge the convolution kernel of the first layer accordingly. The newly introduced matrices are zero-initialized to ensure training stability.

We utilize a relative camera coordinate system, where the first frame is positioned at the world origin with an identity matrix for rotation. The following frames are rotated accordingly. To stabilize training, we normalize the scale of the training sequences to a unit scale. This is implemented by resizing the maximum distance-to-origin in the multi-view camera sequence to 1.

**Cross-frame Attention for Temporal Consistency**    Vanilla 1D temporal attention in the SVD backbone is insufficient for modeling large pixel displacements when the viewpoint changes (Shi et al., 2023b; Yang et al., 2024b). In vanilla 1D temporal attention, attention matrices are calculated over the frame number dimension, and latent features only interact with features from the same spatial location across frames. This limits information flow between different spatial-temporal locations. While this might not be a big issue for video generation with limited motion, viewpoint changes

typically cause significant pixel displacements, which calls for better architecture for more efficient information propagation.

To overcome this issue, we inflate the original 1D temporal attention modules in the SVD network into 3D cross-frame temporal attention modules, allowing for joint modeling of spatial-temporal feature coherence. The inflation operation can be achieved by rearranging the latent features before the attention matrix calculations. Consider the latent features of shape `(B V F C H W)` where `B`, `V`, `F`, `C` refer to the batch size, the number of views, the length of frames, and the feature dimension, respectively, instead of employing 1D attention mechanism on rearranged shape of `((B V H W) F C)`, our inflated attention operates on the rearranged shape `((B V) (F H W) C)`, integrating spatial features into the attention matrices. A visualization is provided in Fig. 1(c).

Since our rearrange operation only alters the sequence length of the attention inputs without modifying the feature dimensions, we can seamlessly inherit the pre-trained weights from the SVD backbone for our purpose. Thanks to this rearrange operation, our inflated temporal attention now calculates the similarity of spatial-temporal features simultaneously, accommodating larger pixel displacements while maintaining temporal consistency.

### 3.3 CONSISTENT MULTI-VIEW VIDEO DIFFUSION MODEL

Adding Plücker coordinates for camera control and introducing improved temporal attention allows the video diffusion model to generate reasonably consistent monocular videos. However, for multi-view generation, a monocular video diffusion model that generates samples independently cannot ensure view consistency across multiple sequences. To address this, we introduce novel design mechanisms and training strategies to extend the monocular video diffusion model to the multi-view generation task.

**Cross-view Attention for Multi-view consistency**  To improve cross-view consistency in multi-view videos, we aim to encourage information exchange during the generation process. Since our temporal cross-frame attention modules already handle intra-view feature connections within each video sequence, we focus on exchanging inter-view signals through the spatial cross-view modules. Inspired by MVDream (Shi et al., 2023b), we introduce 3D cross-view attention modules, inflated from the spatial attention blocks of SVD (Stability, 2023). Specifically, we rearrange the `V` views such that frames at each corresponding timesteps are concatenated before being sent into the attention modules. In detail, we rearrange the latent features from shape `(B V F C H W)` to `((B F) (V H W) C)` instead of `((B V F) (H W) C)`. A visualization is provided in Fig. 1(b).

Since only the second-to-last dimension, representing token length, is extended while other dimensions remain unchanged, our inflated spatial attention can inherit the model weights from the monocular setting. This flexibility allows our model to leverage training data with varying numbers of views and facilitates extrapolation to additional views at inference. To handle multi-view generation, we introduce an additional view dimension to the input data. To maintain workflow simplicity, we absorb the view dimension into the batch dimension during processing of other blocks, ensuring flexibility in handling different numbers of views.

## 4 JOINT TRAINING STRATEGY ON CURATED DATA MIXTURES

Thanks to the view-integrated attention mechanism, which allows for inheriting the module weights with arbitrary viewpoint ($V$) numbers, our framework can leverage various data sources, including static, multi-view dynamic, and monocular videos. This is hard to achieve in previous methods. In this section, we first illustrate our joint training strategy, followed by details on the curated data mixtures that enable this strategy.

### 4.1 JOINT TRAINING STRATEGY FOR VIDEOS WITH VARYING VIEWS

For videos capturing static scenes (Zhou et al., 2018; Yu et al., 2023; Xia et al., 2024; Reizenstein et al., 2021; Deitke et al., 2023b;a), we consider all frames to be temporally synchronized. An arbitrary subsequence of length $(F - 1) \times V + 1$ from the original video can be reformatted into a $V$-view sequence with a shared starting frame and $F$ total frames per view. Static scenes also allow

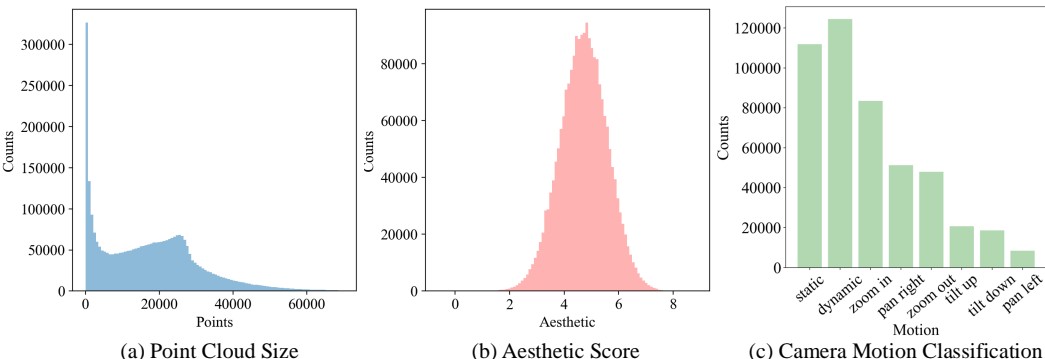

|     (a) Point Cloud Size     |     (b) Aesthetic Score     |     (c) Camera Motion Classification     |

Figure 2: Statistics of our curated monocular video dataset. We provide visualizations of (a) point cloud size, (b) aesthetic score, and (c) camera motion classification result.

frame order reversal, providing additional augmentation opportunities. To overcome the static issue where camera-controlled video models are only able to generate static scenes as in (Wang et al., 2023c; He et al., 2024), we further prepare multi-view dynamic videos by rendering animatable objects from Objaverse (Liang et al., 2024; Jiang et al., 2024). We design random smooth trajectories with diverse elevation and azimuth changes to avoid overfitting on simple camera movements. For each set of renderings, we assign a shared random forward-facing starting point for all $V$ views.

To avoid the model overfitting on synthetic images with simple backgrounds, we include a portion of data from monocular in-the-wild videos (Wang et al., 2023b; Nan et al., 2024). Training multi-view camera control from monocular videos is extremely challenging. Although CamCo (Xu et al., 2024) and 4DiM (Watson et al., 2024) have explored joint training for monocular camera-controllable video generation, these approaches are unsuitable for multi-view scenarios. The concurrent work CVD (Kuang et al., 2024) explored homography warping to augment the monocular videos into pseudo-multi-view videos, but the limited realism of these augmentations restricts their ability to generate complex camera and object motion.

To overcome these issues, we choose to jointly train our model on monocular and multi-view videos to effectively utilize the abundant object motion information from all data sources. We annotate the monocular videos with camera poses using Particle-SfM (Zhao et al., 2022). Since in-the-wild monocular videos often contain noisy or unnatural content, we apply a rigorous filtering pipeline to remove unsuitable clips. These curated video clips, sourced from InternVid (Wang et al., 2023b) and OpenVid (Nan et al., 2024) datasets, provide rich object motion as well as complex backgrounds that mitigate the gap between scene-level static data and object-level dynamic data. We rearrange monocular videos as $V = 1$ samples so that all data items can be consumed by the model in the same way without bells and whistles. Thanks to our view-integrated attention modules, which accommodate varying token lengths introduced by the varying view numbers $V$, the training process remains unaffected when our data items contain both monocular and multi-view videos.

## 4.2 DATA CURATION WORKFLOWS

We begin by training our model extensively on static video data sourced from various publicly available datasets. Wild-RGBD (Xia et al., 2024) includes nearly 20,000 RGB-D videos across 46 common object categories. MVImgNet (Yu et al., 2023) comprises 219,188 videos featuring objects from 238 classes. DL3DV-10K (Ling et al., 2023b) provides 7,000 long-duration videos captured in both indoor and outdoor environments. CO3Dv2 (Reizenstein et al., 2021) contains 34,000 turntable-like videos of rigid objects, crowd-sourced by nonexperts using cellphone cameras. Objaverse (Deitke et al., 2023b) and Objaverse-XL (Deitke et al., 2023a) exhaustively crawl 10 million publicly available 3D assets. From these, we filtered out low-quality assets, such as those with incorrect textures or overly simplistic geometry, yielding a high-quality subset of 400,000 assets.

Similar to Diffusion4D (Liang et al., 2024) and Animate3D (Jiang et al., 2024), we filter the animatable objects from Objaverse's Sketchfab subset. We exclude objects with excessive motion, which might result in partial observations, as well as nearly static objects with minimal motion. This curation process helps us obtain 19,000 high-quality dynamic assets that can be rendered from

arbitrary viewpoints and timesteps, facilitating multi-view video generation. During each training iteration, we augment the frames with randomly selected background colors.

To improve the model's ability to generate object motion in the presence of complex backgrounds, we prepare monocular videos with camera pose annotations similar to CamCo (Xu et al., 2024). First, we use Particle-SfM (Zhao et al., 2022) to estimate the camera poses for randomly sampled frames from videos from InternVid (Wang et al., 2023b) and OpenVid (Nan et al., 2024). Inspired by CO3D (Reizenstein et al., 2021) and CamCo (Xu et al., 2024), we remove the videos where SfM fails to register all available frames or produces a point cloud with too few points or too many points. Fig. 2(a) shows the point count statistics. A point cloud with too few points indicates poor frame registration to a shared 3D representation, while too many points suggest a mostly static scene, which is undesirable as we focus on object motion. Additionally,

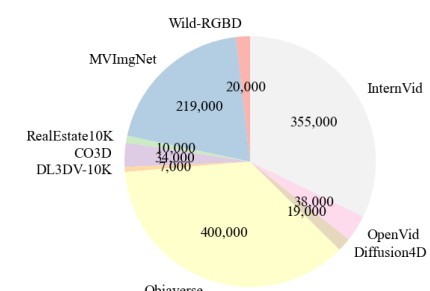

Figure 3: Sources of our training videos. We visualize the numbers of raw videos from each source. During training, we perform augmentations to the raw video sequences to avoid overfitting on certain camera motions.

non-registered frames may indicate potential scene changes. We then apply a rigorous filtering pipeline to ensure the quality of the video samples used for training. This includes filtering based on aesthetic scores, optical character recognition (OCR), and camera motion classification using optical flow. Videos containing detected character regions are aggressively removed. Fig. 2(b) and (c) present statistics on aesthetic score and camera motion classification results. Videos with low aesthetic scores or those classified as having static camera motion are excluded from the training set. Ultimately, we construct a dataset of 393,000 monocular videos annotated with camera poses. We provide a summary of the data sources used in Fig. 3. More details and analysis are provided in the appendix.

## 5 EXPERIMENTS

In this section, we present experimental results and analysis. Video comparisons are included in the supplementary material for optimal visual evaluation. It is important to note that for all qualitative and quantitative evaluations, neither the input images nor the camera trajectories were used during model training.

### 5.1 QUANTITATIVE COMPARISONS

**3D Consistency of Frames** We evaluate the 3D consistency of the generated videos using COLMAP (Schönberger & Frahm, 2016; Schönberger et al., 2016). COLMAP is widely adopted for 3D reconstruction methods where camera pose estimation is required for in-the-wild images. We configure the COLMAP following previous methods (Deng et al., 2022; Xu et al., 2024) for best few-view performance. A higher COLMAP error rate indicates poorer 3D consistency in the input images. Motivated by this, we report COLMAP errors as a measure of the 3D consistency of the frames. Each video is retried up to five times to reduce randomness. We randomly sample 1,000 video sequences from RealEstate10K (Zhou et al., 2018) test set for evaluation. Since we have ground truth 3D scenes, we use the ground truth camera pose sequences as the viewpoint instruction of the video model and compare the generated frames against the ground truth images. Similar to prior works (He et al., 2024; Xu et al., 2024), we extract the estimated camera poses and calculate the relative translation and rotation differences. Specifically, given two camera pose sequences, we convert them to relative poses and align the first frames to world origin. We then measure the angular errors in translation and rotation. Unlike previous works (He et al., 2024; Xu et al., 2024) that calculate the Euclidean distance of translation vectors, we use angular error measurements to ensure the camera pose scales are normalized, addressing scale ambiguity. As shown in Tab. 1, we calculate the area under the cumulative error curve (AUC) of frames whose rotation and translations are below certain thresholds ($5°$, $10°$, $20°$). Our method significantly outperforms existing baselines.

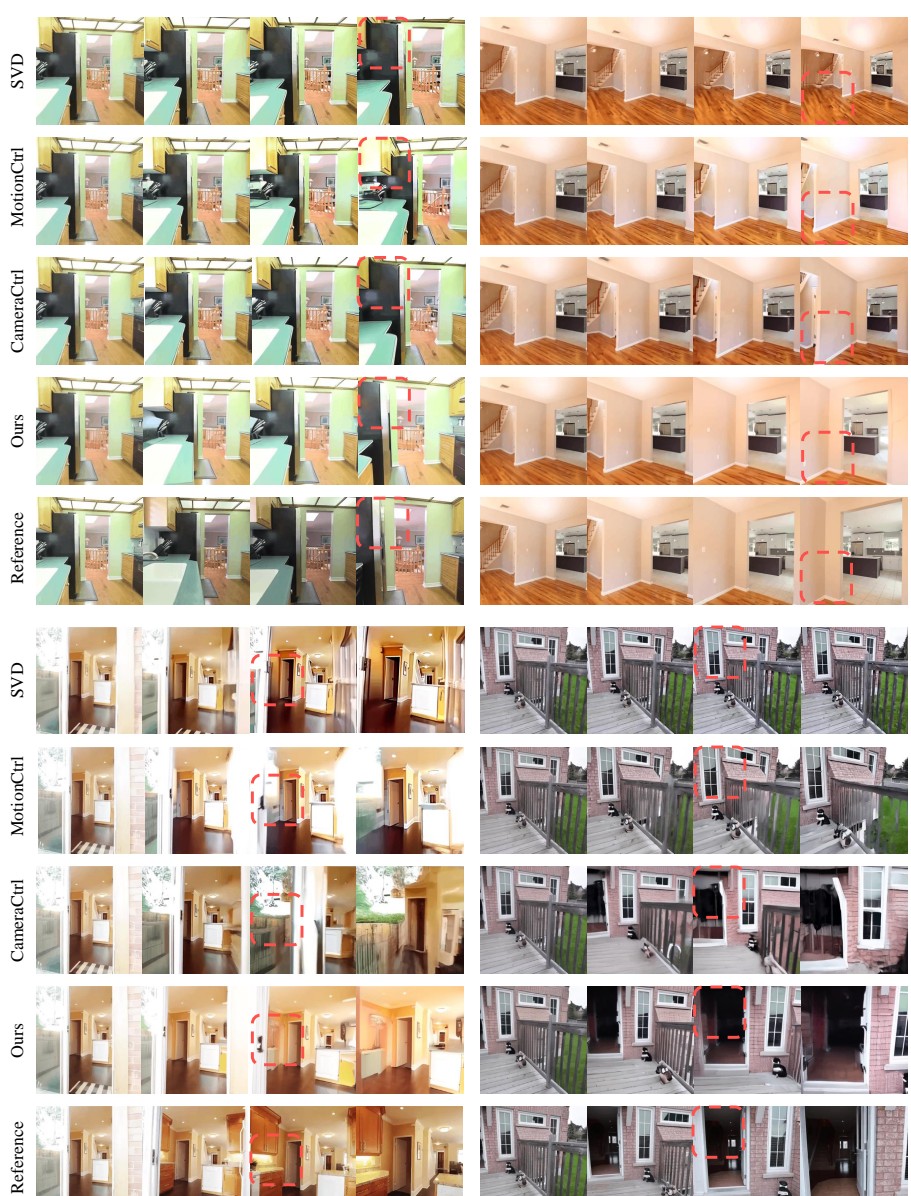

Figure 4: Per-video qualitative comparisons. The first frame in each reference set is the input image. Neither the image nor the camera trajectories were seen during model training. Video results are provided in supplementary for clearer qualitative comparisons.

Table 1: Quantitative comparison for monocular geometry consistency on RealEstate10K test set.

| Methods | FID ↓ | FVD ↓ | COLMAP error↓ | Rot. AUC ↑ (@5°/10°/20°) | Trans. AUC ↑ (@5°/10°/20°) |
|---|---|---|---|---|---|
| SVD | 16.89 | 139.64 | 30.3% | 14.4 / 22.8 / 35.3 | 0.2 / 1.0 / 3.2 |
| MotionCtrl | 21.09 | 119.06 | 55.0% | 8.6 / 13.9 / 22.2 | 0.6 / 2.1 / 5.7 |
| CameraCtrl | 14.69 | 105.41 | 19.3% | 21.4 / 32.9 / 48.4 | 0.3 / 1.3 / 4.4 |
| Ours | **11.43** | **55.10** | **14.4%** | **22.9 / 34.5 / 50.1** | **5.1 / 12.7 / 24.6** |

**Multi-view Consistency**    Alongside evaluating the individual monocular frame pose accuracy using COLMAP-based metrics, we further assess the cross-video consistency of the corresponding frames from generated multi-view videos. We randomly sample 1,000 videos, each with 27 frames, from RealEstate10k (Zhou et al., 2018) test set and convert each video into a two-view sequence with 14 frames per view. The new camera pose sequences are generated by setting the 14th frame as the world origin and positioning the remaining frames relative to it. The scales of the scenes are normalized

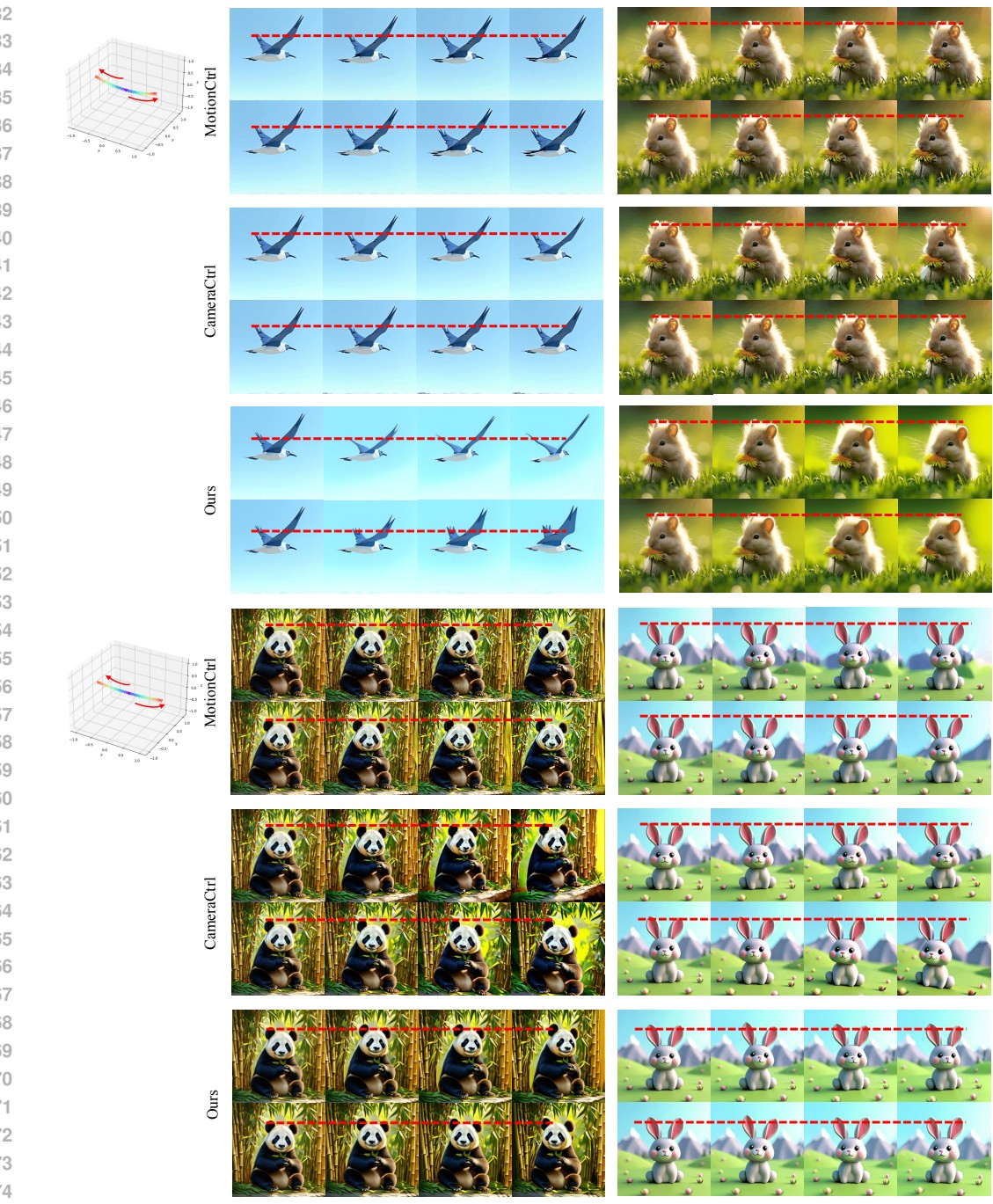

Figure 5: Qualitative comparisons for 2-view video generations. Each generation consists of two rows, where each row represents a sequence of generated frames, with columns showing frames at the same timestep. Neither the image nor the camera trajectories were used during model training. Red dotted lines are annotated to highlight object motion. Video results are included in the supplementary material for clearer comparisons.

so that the maximum distance from the origin is 1. Following CVD (Kuang et al., 2024), we adopt SuperGlue (Sarlin et al., 2020) to find correspondences and estimate the camera poses between each time-aligned set of frames. SuperGlue not only measures angular errors in the rotation and translation but also computes the epipolar error of the matched correspondences. We similarly collect the AUC for frame pairs with rotation and translation errors below specific thresholds ($5°$, $10°$, $20°$). The

Table 2: Quantitative comparison for 2-view video generation.

| Scenes | Methods | FID ↓ | FVD ↓ | Rot. AUC ↑ (@5°/10°/20°) | Trans. AUC ↑ (@5°/10°/20°) | Prec. ↑ | MS. ↑ |
|--------|---------|-------|-------|-----------------|------------------|---------|-------|
| Real10K | SVD | 37.99 | 296.95 | 7.9 / 13.5 / 28.2 | 0.2 / 0.7 / 2.4 | 6.49 | 4.17 |
| | MotionCtrl | 29.23 | 277.05 | 8.1 / 16.5 / 29.4 | 1.5 / 5.3 / 16.1 | 11.45 | 5.90 |
| | CameraCtrl | 12.57 | 131.32 | 22.4 / **38.5** / **56.2** | 0.6 / 2.5 / 8.2 | 19.49 | 11.25 |
| | Ours | **8.82** | **94.86** | **23.9** / 37.4 / 52.9 | **3.3 / 10.2 / 23.5** | **29.39** | **15.22** |
| General | MotionCtrl | 47.31 | 313.92 | 4.9 / 11.3 / 21.9 | 0.7 / 2.4 / 8.2 | 8.12 | 3.93 |
| | CameraCtrl | 26.71 | 221.23 | 14.1 / 26.9 / 43.2 | 0.5 / 1.7 / 5.7 | 15.13 | 7.35 |
| | Ours | **26.12** | **173.70** | **19.7 / 32.7 / 48.4** | **0.8 / 2.8 / 8.7** | **33.10** | **19.96** |

epipolar errors for the estimated correspondences are summarized to the precision (P) and matching score (MS). As shown in Tab. 2, our method outperforms baselines greatly. The "Real10K" category means that the input images are taken from the corresponding RealEstate10K test sequence, while the "General" means that the input images are taken from 1,000 randomly sampled images in the test split of our monocular video dataset.

**Visual Quality**    To assess the frame perceptual quality, we evaluate visual quality using FID (Heusel et al., 2017) and FVD (Unterthiner et al., 2018). FID and FVD measure the feature-space similarity of two sets of images and videos, respectively. In our case, they quantify the distribution distance between the generated frame sequences and the ground-truth frames. We provide monocular evaluations in Tab. 1 and multi-view evaluations in Tab. 2. As shown in these tables, our proposed framework enjoys the best visual quality. For both the "Real10K" and "General" categories, the ground-truth videos used to calculate these metrics are the video sequences corresponding to the input frames. These video sequences are from the test set split of the datasets and are not seen during training.

## 5.2 QUALTITATIVE COMPARISON

We provide qualitative comparisons on RealEstate10k (Zhou et al., 2018) scenes in Fig. 4 and text-to-image generated images in Fig. 5. As shown in Fig. 4, our method produces videos with precise camera control, whereas MotionCtrl tends to generate overly smooth trajectories that simplify the viewpoint instructions, and CameraCtrl suffers from severe distortions at novel viewpoints. For example, in the first case, the camera instruction involves multiple panning operations, first panning left and then panning right. Still, MotionCtrl only pans left, ignoring the rest of the instructions. CameraCtrl's outputs, particularly in the first two cases, exhibit noticeable distortion, with the walls bending in the later frames. Additionally, in the third and fourth cases, where the camera trajectories cover a long distance, both MotionCtrl and CameraCtrl produce unrealistic hallucinations, introducing artifacts such as merging indoor and outdoor pixels or distorting input pixels to compensate for a lack of generation ability. In Fig. 5, we observe that MotionCtrl and CameraCtrl tend to generate static scenes without any object motion. Although their methods produce realistic novel views, the synthesized objects remain static. In contrast, our method generates vivid object motion while maintaining accurate camera control. We highlight the object motion in Fig. 5 using auxiliary red lines. We encourage readers to view the supplementary videos for optimal visual comparisons.

## 5.3 ABLATION STUDIES AND APPLICATIONS

Due to the space limit, we refer readers to the Appendix for ablation studies and applications of our framework. We provide detailed ablation studies in Sec. D on our proposed framework. Additionally, we explore the 3D reconstruction of our generated frames and four-view generation capabilities in Sec. E. Videos are included in the supplementary material for optimal qualitative comparison.

## 6 CONCLUSION

In this paper, we propose Cavia, a novel framework for consistent multi-view camera-controllable video generation. Our framework incorporates cross-frame and cross-view attentions for effective camera controllability and view consistency. Our model benefits from joint training using static 3D scenes and objects, animatable objects, and in-the-wild monocular videos. Extensive experiments demonstrate the superiority of our method over previous works in terms of geometric consistency and perceptual quality.

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

## A    ADDITIONAL IMPLEMENTATION DETAILS

Our training is divided into static stage and dynamic stage. Our static stage is trained for around 500k iterations and our dynamic stage is trained for roughly 300k iterations. The effective batch size is 128 and the learning rate is 1e-4. Our video length is 14 frames for each view with the first frame shared across views. Our model is fine-tuned at $256 \times 256$ spatial resolution from the SVD 1.0 checkpoint. The training data are prepared by first center-cropping the original videos and then resizing each frame to the shape of $256 \times 256$. In the dynamic stage, 30% of iterations are used to train on monocular videos. During static training, the strides of frames are randomly sampled in the range of `[1, 8]`. For monocular videos, the strides are sampled in the range of `[1, 2]`. For dynamic multi-view object renderings, the strides are fixed to 1 to use all rendered frames since we already introduced randomness in the frame rate during rendering. At inference time, the decoding chunk is set to 14 so all frames are decoded altogether. We sample 25 steps to obtain all our results.

## B    ADDITIONAL DATA CURATION DETAILS

In this section, we provide additional details on our data processing and curation pipelines.

**Static 3D Objects**    Our static objects data comprises multi-view images rendered from the Objaverse (Deitke et al., 2023b) and Objaverse-XL(Deitke et al., 2023a) dataset. Similar to InstantMesh, we use a filtered high-quality subset of the original dataset to train our model. The filtering goal is to remove objects that satisfy any of the following criteria: (i) objects without texture maps, (ii) objects with rendered images occupying less than 10% of the view from any angle, (iii) including multiple separate objects, (iv) objects with no caption information provided by the Cap3D dataset, and (v) low-quality objects. The classification of "low-quality" objects is determined based on the presence of tags such as "lowpoly" and its variants (e.g., "low poly") in the metadata. By applying our filtering criteria, we curated approximately 400k high-quality instances from the initial pool of 800k objects in the Objaverse dataset.

For each 3D object, we use Blender's EEVEE renderer to render an 84-frame RGBA orbit at $512 \times 512$ resolution. we adaptively position the camera to a distance sufficient to ensure that the rendered object content makes good and consistent use of the image extents without being clipped in any view. For each frame, the azimuths can be irregularly spaced, and the elevation can vary per view. Specifically, the sequence of camera elevations for each orbit is obtained from a random weighted combination of sinusoids with different frequencies. The azimuth angles are sampled regularly, and then a small amount of noise is added to make them irregular. The elevation values are smoothed using a simple convolution kernel and then clamped to a maximum elevation of 89 degrees.

**Static 3D Scenes**    Our static scenes data are sourced from RealEstate10k (Zhou et al., 2018), WildRGBD (Xia et al., 2024), MVImgNet (Yu et al., 2023), CO3Dv2 (Reizenstein et al., 2021), and DL3DV-10K (Ling et al., 2023b). For RealEstate10k, we use the train/test split released by PixelSplat (Charatan et al., 2023). During training, we sample every 8 original frames to construct the training sequences. For DL3DV-10K, we construct training sequences from the publicly available 7k subset. Since each video is very long for the DL3DV-10k dataset, we offline randomly sample multiple sequences from a single ground truth video to obtain multiple training data items. For CO3Dv2, we remove the video sequences that contain whole-black images to avoid temporally inconsistent frames. For WildRGBD and MVImgNet we use all classes available and removed sequences whose lengths are not enough for two-view training (shorter than 27 frames).

**Dynamic 3D Objects**    Our dynamic 3D objects are similarly rendered as the static 3D objects. The filtering pipelines remain mostly the same as the static objects, except that we introduce additional workflows to consider object motion. Inspired by previous works (Liang et al., 2024; Jiang et al., 2024; Li et al., 2024a) that employ animatable objects from Objaverse. We render multiple fixed-view videos to examine the motion quality of the objects. We utilize lpips (Zhang et al., 2018) to measure the similarity of nearby frames and consider an object to be static if lpips similarity is above a certain threshold. Additionally, we render the alpha masks of the object and use this as an indicator of whether the object has moved out of the visible regions. Consequently, we remove objects with too large or sudden movements as well as objects with little-to-no motion. These filterings result in

Table 3: Ablation Studies on each of our introduced modules. "w/o Plücker" refers to replacing the Plücker coordinate conditioning with one-dimensional conditioning as in MotionCtrl. "w/o Cross-frame" refers to replacing the Cross-frame attention with vanilla 1D temporal attention. "w/o Cross-view" refers to replacing the Cross-view attention with vanilla spatial attention. "Ours (Static)" means the model is only trained on static video datasets. "Ours (w/o Mono)" means that the model is fine-tuned on synthetic multi-view datasets, but is not trained with monocular video datasets. "Ours (Full)" means that the model is trained on all available data sources.

| Scenes | Methods | FID↓ | FVD↓ | Rot. AUC ↑ (@5°/10°/20°) | Trans. AUC ↑ (@5°/10°/20°) | Prec. ↑ | MS. ↑ |
|---|---|---|---|---|---|---|---|
| Real10K | w/o Plücker | 12.75 | 195.84 | 12.1 / 21.9 / 35.5 | 1.6 / 5.8 / 16.4 | 14.74 | 10.02 |
| | w/o Cross-frame | 17.04 | 154.54 | 21.4 / 34.8 / 50.1 | 3.8 / 11.1 / 24.2 | 25.67 | 12.70 |
| | w/o Cross-view | 9.45 | 106.82 | 22.8 / 36.7 / 52.4 | 2.7 / 8.7 / 22.1 | 27.57 | 14.65 |
| | Ours | **8.82** | **94.86** | **23.9 / 37.4 / 52.9** | **3.3 / 10.2 / 23.5** | **29.39** | **15.22** |
| General | w/o Cross-frame | 71.39 | 249.02 | 9.8 / 19.1 / 32.7 | 0.5 / 1.9 / 6.6 | 13.20 | 8.97 |
| | w/o Cross-view | 30.89 | 246.68 | 14.9 / 27.4 / 42.9 | 1.2 / 4.3 / 12.2 | 17.58 | 9.59 |
| | Ours (Static) | 27.20 | 185.58 | 15.9 / 28.7 / 44.1 | 1.4 / 4.6 / 12.9 | 21.75 | 12.04 |
| | Ours (w/o Mono) | 35.79 | 243.05 | 15.0 / 27.1 / 42.6 | 0.3 / 1.3 / 4.2 | 18.55 | 10.78 |
| | Ours (Full) | **26.12** | **173.70** | **19.7 / 32.7 / 48.4** | **0.8 / 2.8 / 8.7** | **33.10** | **19.96** |

19,000 objects. Our rendering strategy is also very similar to that of static 3D objects, introducing random elevation and azimuth changes to complicate the trajectories, except that we additionally introduce a random frame stride at rendering to augment the object motion. The stride is sampled individually for each object from the range `[1, 3]`. A larger the stride leads to renderings with faster object motion.

**Monocular Videos**  Our monocular video filtering pipeline involves filtering according to Particle-SfM output, OCR, aesthetic score, and camera motion. As mentioned in Sec. 4.2, we first attempt to annotate the camera poses for the video frames using Particle-SfM (Zhao et al., 2022). Take InternVid (Wang et al., 2023b) as an example, roughly 10 million video clips are processed and around 3 million samples are successfully processed by Particle-SfM. For each video, we start from the first frame and randomly select a frame stride of 1 or 2. The total number of images sent to Particle-SfM is 32 images. Our point count filtering is empirically implemented as a cut-off at 1,000 points and 40,000 points. Point clouds with too few points are removed due to the concern that the frames are poorly registered. Point clouds with too many points are avoided because their limited object motion. This aggressive filtering results in around 2 million samples for further processing. We then evaluate all the video clips using OCR detection algorithms and remove the samples whose detected text regions are larger then $10^{-4}$ of the image resolution (*i.e.* 6 pixels). This process results in 604,000 samples. The next step is filtering with aesthetic scores and videos with aesthetic score annotations smaller than 4 are removed. 467,000 videos are left after these filtering process. Finally, we employ a camera motion classifier extended from the Open-Sora pipeline[1]. The main motivation is that optical-flow on consecutive frames can be summarized to a global motion vector, assuming the most parts of the scene is moving in a uniform direction. Optical flow is first obtained using `cv2.calcOpticalFlowFarneback` for each consecutive frame pairs. Then, the magnitudes and directions are calculated via `cv2.cartToPolar`. These magnitudes and directions are classified into 8 categories: static, zoom out, zoom in, pan left, tilt up, pan right, tilt down, and unknown. The results of the frame pairs are summarized to obtain the final result of each video clip. When a certain type appears more than 50%, the type for the whole video clip is determined directly. We aggressively classify a video clip as static if any of its frame pairs is categorized into static or unknown. Finally, we obtain 355,000 clips that satisfy our needs. The process is similarly applied to OpenVid (Nan et al., 2024)'s Panda-70M subset Chen et al. (2024a) and we obtained 38,000 clips. In summary, our monocular video dataset consists of 393,000 clips.

---

[1] https://github.com/hpcaitech/Open-Sora/tree/main/tools/caption/camera_motion

| Model | Plücker Coordinates | Cross-frame Attention | Cross-view Attention |
|---|:---:|:---:|:---:|
| w/o Plücker | ✗ | ✗ | ✗ |
| w/o Cross-frame | ✓ | ✗ | ✗ |
| w/o Cross-view | ✓ | ✓ | ✗ |
| Ours | ✓ | ✓ | ✓ |

Table 4: Illustration of the model variants in the ablation studies.

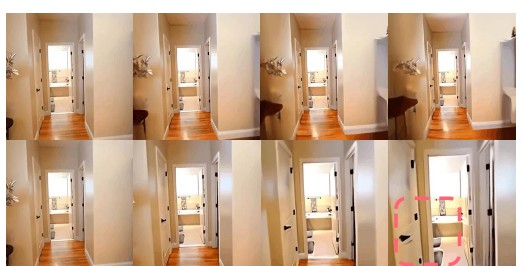 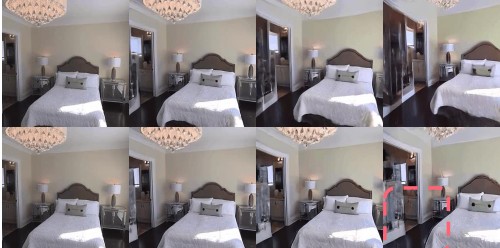

(a) Without Plücker

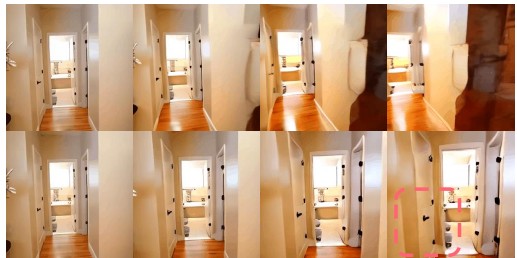 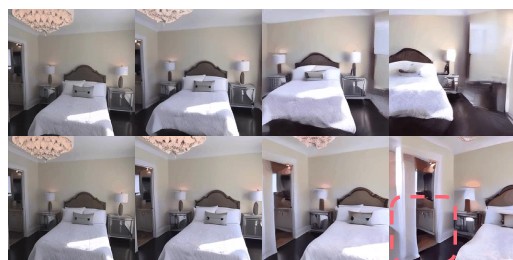

(b) Without Cross-frame Attention

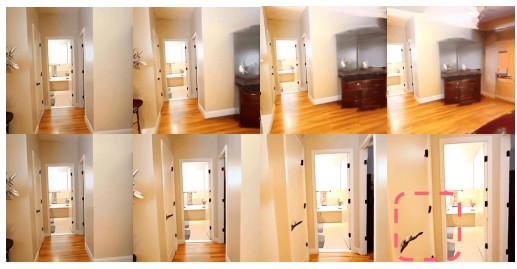 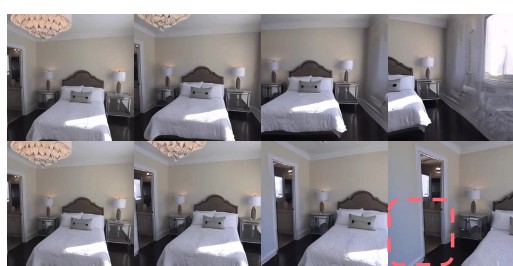

(c) Full Model

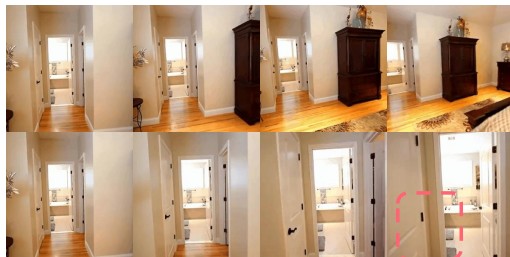 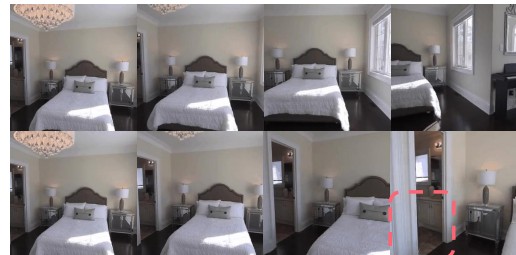

(d) Reference

Figure 6: Ablation studies on Plücker coordinates and Cross-frame Attention. Video results are provided in supplementary for clearer qualitative comparisons.

## C  EVALUATION DETAILS

For MotionCtrl and CameraCtrl, we use the open-source checkpoints trained from SVD released by the authors. These checkpoints are designed for image-to-video tasks so we can have fair compar-

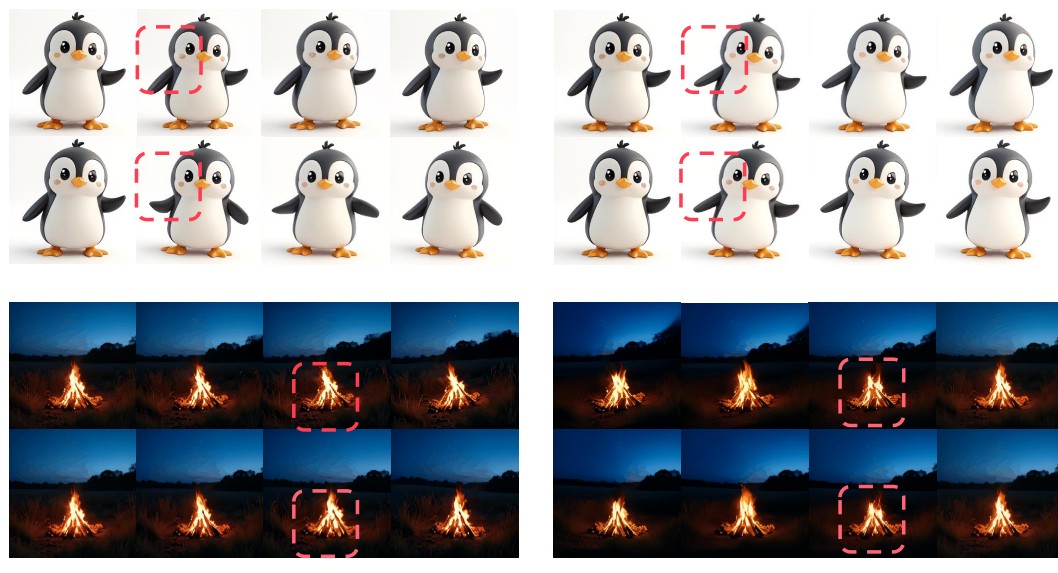

(a) Without Cross-view Attention          (b) With Cross-view Attention

Figure 7: Ablation studies on Cross-view Attention. Video results are provided in supplementary for clearer qualitative comparisons.

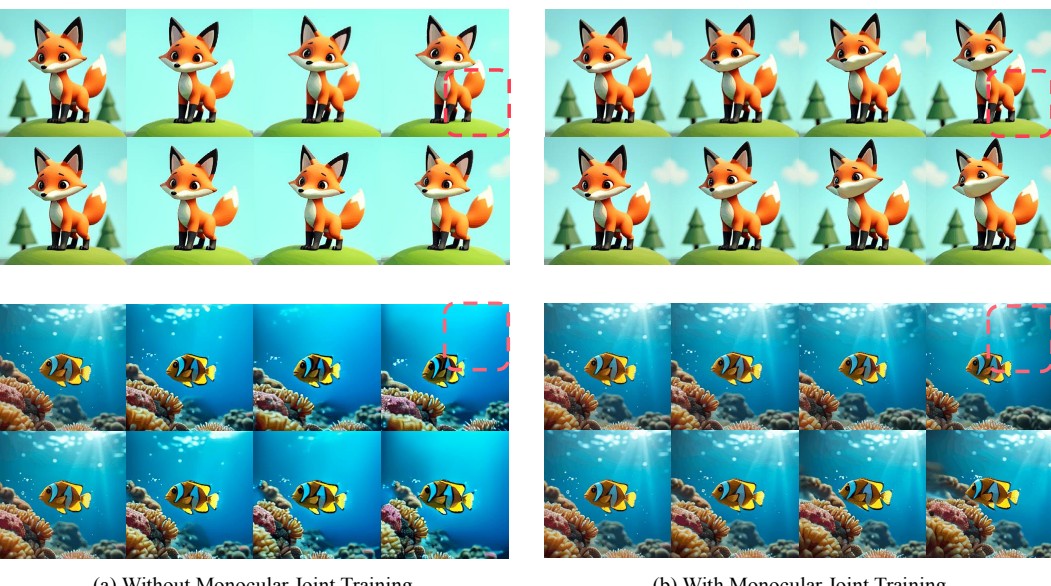

(a) Without Monocular Joint Training         (b) With Monocular Joint Training

Figure 8: Ablation studies on the joint training strategy on monocular videos. Video results are provided in supplementary for clearer qualitative comparisons.

isons. We use "clean-fid"[2] and "common-metrics-on-video-quality"[3] for obtaining FID and FVD, respectively. Our FVD results are reported in VideoGPT (Yan et al., 2021) format. Our COLMAP is configured following DSNeRF (Deng et al., 2022) and (Xu et al., 2024) to improve the few-view reconstruction performance. Concretely speaking, we enable `--SiftMatching.max_num_matches 65536` to support robust feature matching. To ensure that the SfM results best align with our videos, we set `--ImageReader.single_camera 1` since most videos in our datasets consist of frames captured from a single camera.

---

[2]`https://github.com/GaParmar/clean-fid`
[3]`https://github.com/JunyaoHu/common_metrics_on_video_quality`

## D    ABLATION STUDIES

In this section, we conduct extensive evaluations for ablation studies. We provide video comparisons in the supplementary. We provide thorough quantitative comparisons in Tab. 3 to illustrate the importance of our proposed components. In Tab. 4, we illustrate the differences between model variants used for ablation studies. The models are evaluated using RealEstate10K camera trajectories. For the "Real10K" and "General" categories, the testing images are from our test set split of RealEstate10K and InternVid, respectively. Our full model enjoys the best perceptual quality and geometric consistency.

We first examine the importance of Plücker coordinates conditioning and the cross-frame attention modules. As shown in Fig. 6, model variants without cross-frame attention contains severe distortion artifacts, such as the bent walls. The model variant without Plücker coordinates results in simplified camera motion that ignores the complex camera viewpoint instructions.

We then evaluate the model variant without cross-view attention. As shown in Fig. 7, we observe that removing the cross-view attention module results in multiple individual video samples that contain different object motions. For example, the penguin moves differently in the first case, and the wood sticks in the fire appear differently in the second case. This behavior is not desirable because our goal is to obtain multiple videos from different camera paths of the same scene.

Finally, we examine the importance of our monocular video joint training strategy. As shown in Fig. 8, we observe that when overfitting on dynamic objects from Objaverse, the generated results tend to contain frames with simplified backgrounds. This is mainly because, during the training, all data samples from Objaverse are implemented with single random color backgrounds. Our model benefits from joint training on monocular videos and preserves the ability to generate complex backgrounds when object motion is present.

## E    APPLICATIONS

In this section, we provide additional results on four-view inference and 3D reconstruction of our generated frames.

### E.1    ADVANCING TO FOUR VIEWS AT INFERENCE

Our cross-view attention design enables us to extrapolate to more views straightforwardly at inference time. This design is more efficient compared with the concurrent work CVD (Kuang et al., 2024) which requires enumeration of viewpoint pairs at inference time. We conduct a side-by-side comparison for 4-view generation in Fig. 9. Our method enjoys better consistency and shows more realistic results than CVD (Kuang et al., 2024). In comparison, CVD tends to produce artifacts at border regions. For example, the structure of the wall (first case) and the window (second case) change when the viewpoint changes. The results from CVD are taken from their author's website. We provide video comparisons in the supplementary. We also provide more 4-view generation results from Cavia in our supplementary.

### E.2    3D RECONSTRUCTION OF GENERATED FRAMES

We further perform 3D reconstruction on our generated frames. We render our reconstructed 3D Gaussians from an elliptical trajectory consisting of 16 novel views. We provide a side-by-side comparison with the concurrent work CVD (Kuang et al., 2024) in Fig. 10. Compared with the results of CVD, our frames are more geometrically consistent and result in clearer 3D reconstruction and fewer floaters. For example, the results from CVD produce floaters on the cupboard regions and generate blurry artifacts for the wall and the TV due to inconsistencies. We provide video comparisons in the supplementary for clearer comparisons. We also provide additional 3D reconstruction results of Cavia's generated frames in the supplementary.

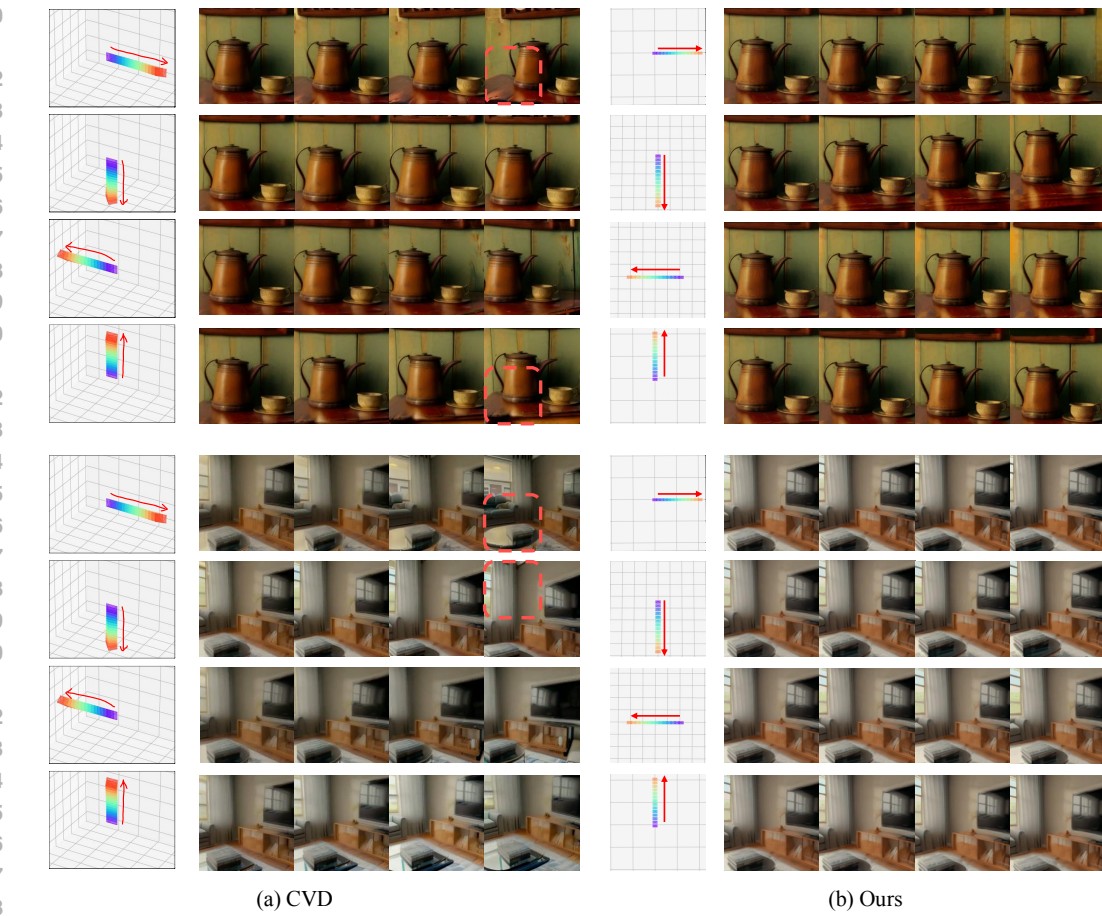

(a) CVD                                   (b) Ours

Figure 9: Four-view video comparison. The result of CVD is taken from their website. CVD tends to generate black border pixels, potentially due to its homography warping augmentations during training. In comparison, our method produces frames with better geometric consistency and perceptual quality.

## F  LIMITATIONS

Our framework has limited ability to generate large object motion, mainly due to the limitation of the base video generator SVD (Stability, 2023). We will explore better base models in future works. Moreover, our data curation pipelines assume a simple camera model using shared camera intrinsic across frames. While enabling easier data preparation, this limits our model from generalizing to complex camera intrinsic changes at inference time, which is widely adopted in cinematography. Additionally, for simplicity, our framework is trained with normalized scales of scenes, which can be further improved if potentially calibrated with metric scale. We will explore calibration techniques for better quality if a well-generalizable metric depth estimator becomes publicly available.

## G  ADDITIONAL DYNAMIC RESULTS

We provide additional dynamic results as suggested by reviewers in Fig. 11. The gif version are submitted with the original supplementary webpage.

## H  DATA SAMPLES

We provide visualizations for the annotations from Particle-SfM in Fig. 12. The good one is an example of a high-quality pose estimation result, while the bad one is a failure example.

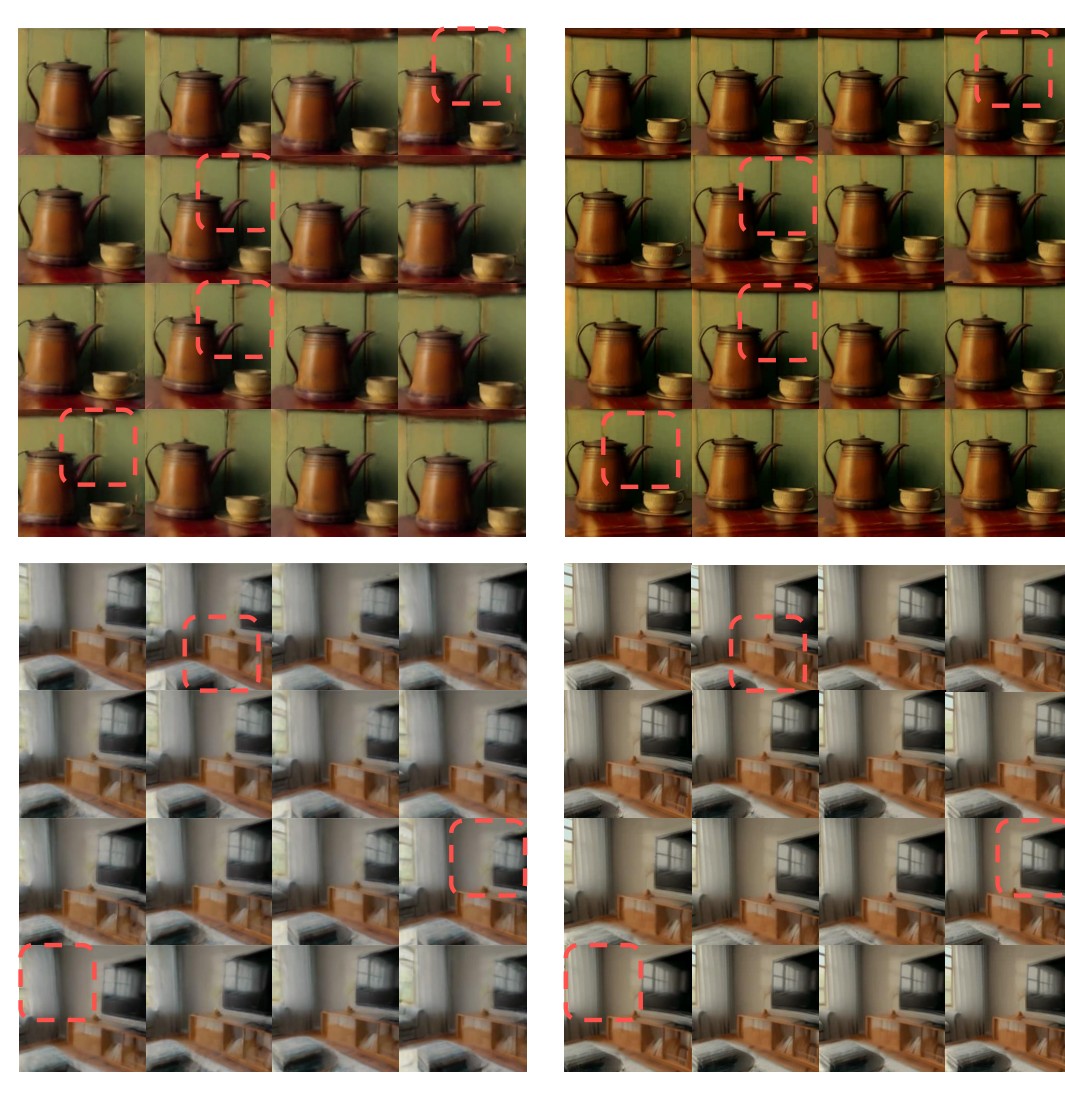

(a) CVD  (b) Ours

Figure 10: 3D Reconstruction comparison. We render the reconstructed 3D Gaussians from an elliptical trajectory consisting of 16 novel views. The result of CVD is taken from their website. CVD's reconstruction results suffer from floaters and blurry artifacts due to the inconsistency in their generated frames. In comparison, our method produces sharper results with clearer visual quality.

We also provide examples of each data source in Fig. 13. Thanks to our joint training strategy, our model supports data samples with arbitrary viewpoint numbers. As a result, our model is jointly trained on static scenes, dynamic objects, and monocular videos.

## I   MOTION COMPARISON WITH SVD

As suggested by the reviewers, we provide side-by-side comparisons with our base model (SVD). As shown in Fig. 14, the frames generated by our method exhibit rich object motion that is comparable to, if not better than, that of SVD. For example, our penguin, giraffe, and bird examples are behaving realistically, while SVD presents limited object motion.

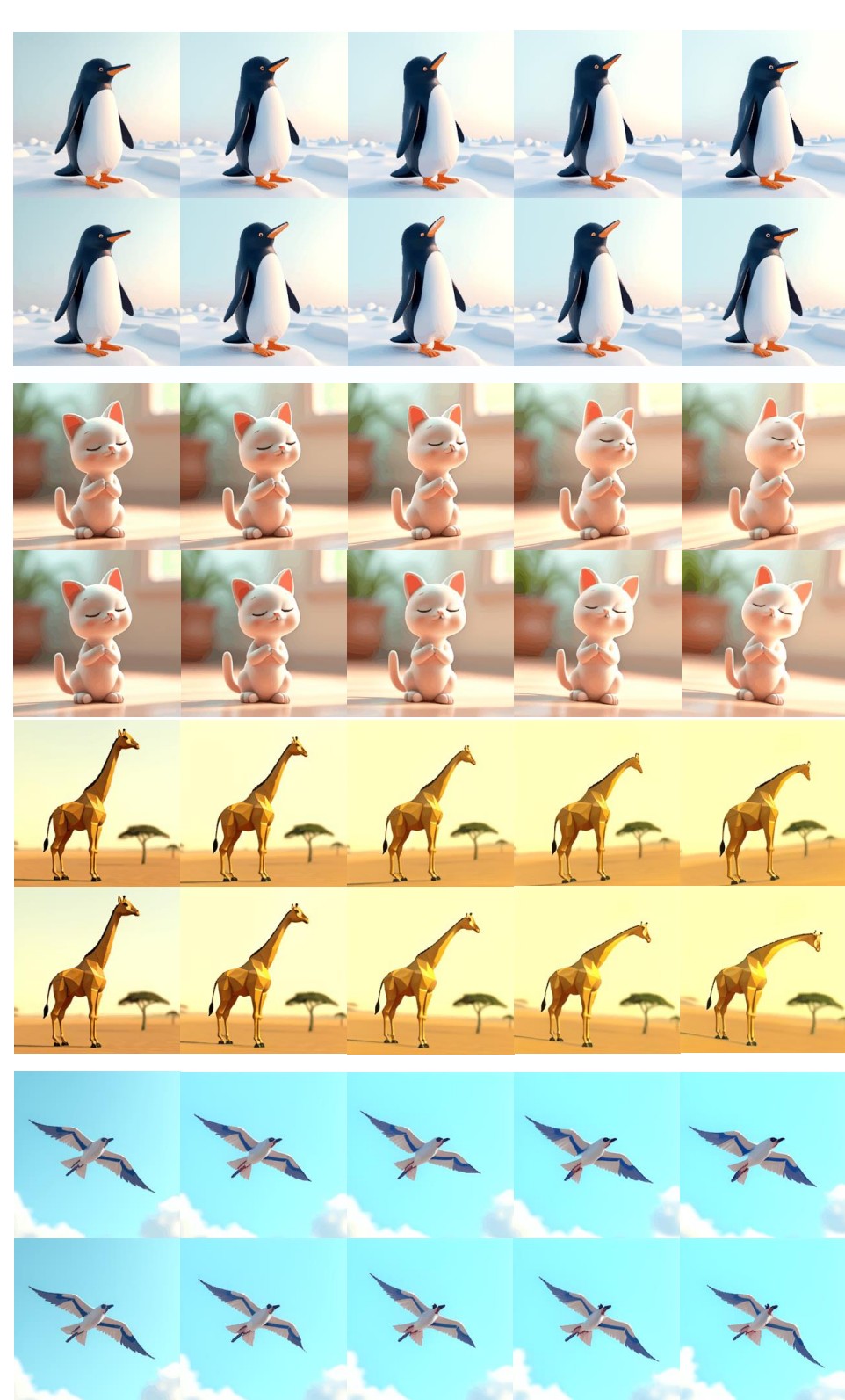

Figure 11: We provide additional dynamic results as suggested by reviewers. The gif version are submitted with the original supplementary webpage.

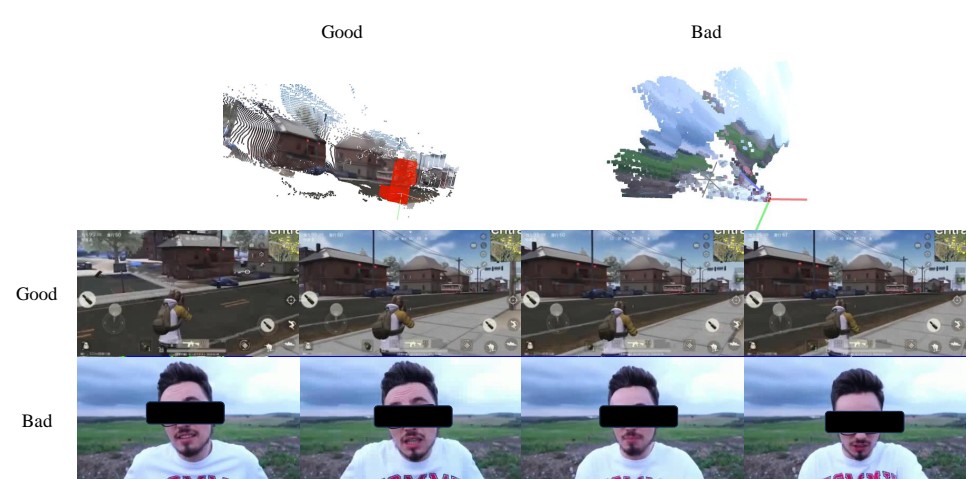

Figure 12: Examples of the annotations produced by Particle-SfM. We add black boxes to the original frames to avoid revealing the identity.

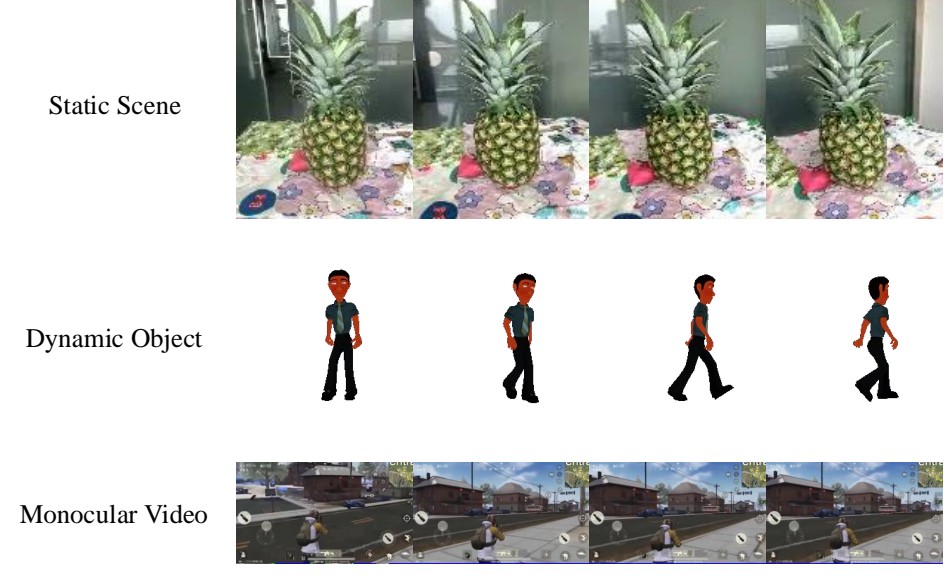

Figure 13: Examples of our data samples from various sources.

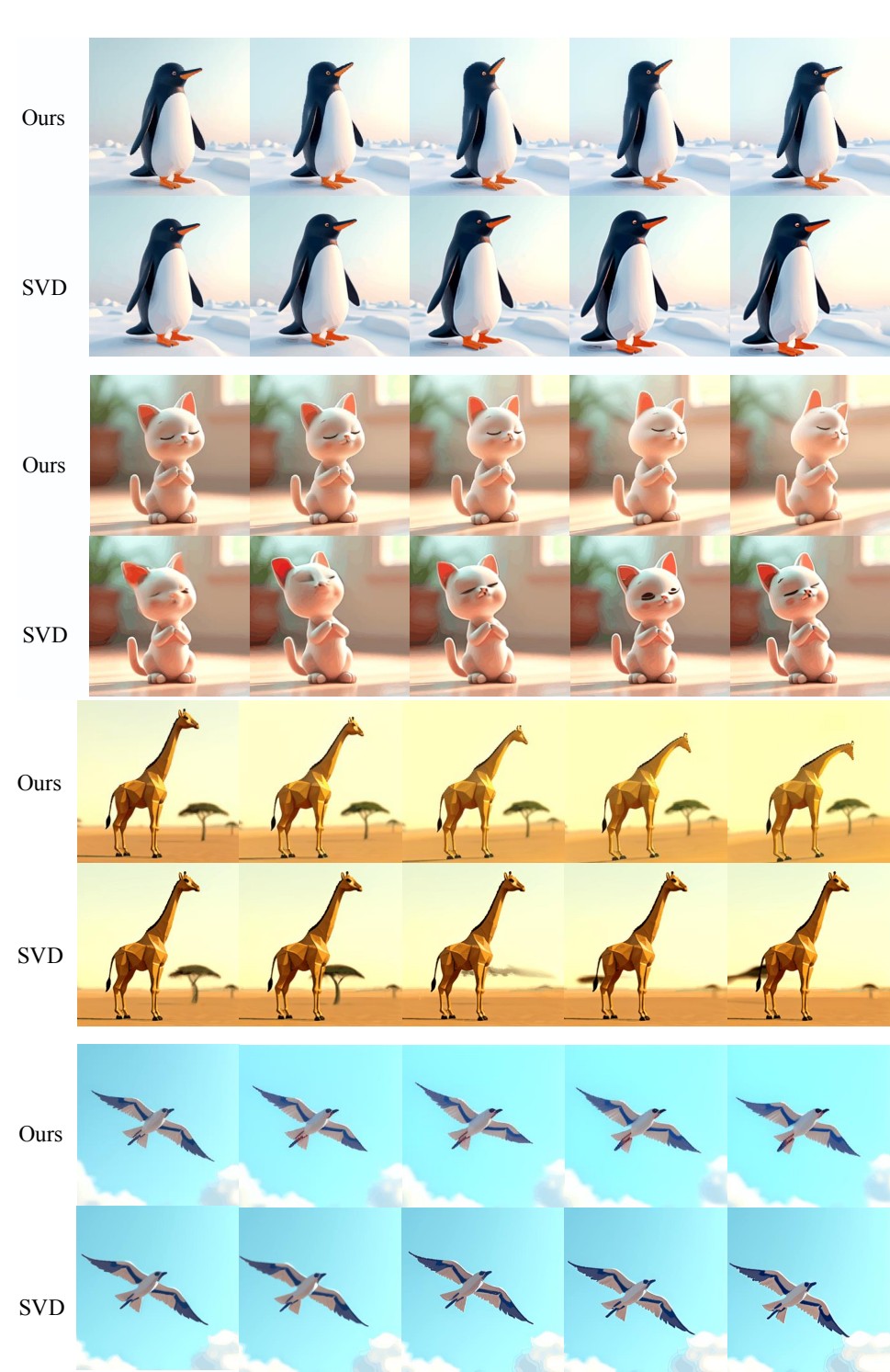

Figure 14: We compare the object motion against our base model (SVD). The frames generated by our method exhibit rich object motion that is comparable to, if not better than, that of SVD. For example, our penguin, giraffe, and bird examples are behaving realistically, while SVD presents limited object motion.

