# OpenReview forum: "Cavia: Camera-controllable Multi-view Video Diffusion with View-Integrated Attention"
_ICLR.cc/2025/Conference — Submitted to ICLR 2025_

### Official Review · Reviewer_UoYX · 2024-10-16

**Soundness:** 3
**Presentation:** 3
**Contribution:** 2
**Rating:** 5
**Confidence:** 5

**Summary:**

This paper introduces Cavia, a novel framework for generating multi-view videos with camera controllability. It incorporates view-integrated attention mechanisms—specifically, cross-view and cross-frame 3D attentions—to enhance consistency across different viewpoints and frames. The paper also presents an effective joint training strategy that leverages a curated mix of static, monocular dynamic, and multi-view dynamic videos. This approach ensures geometric consistency, high-quality object motion, and background preservation in the generated results. Experimental results show that Cavia outperforms baseline methods in terms of geometric and perceptual quality for monocular video generation and cross-video consistency. Additionally, the flexible framework can process four views at inference, leading to improved view consistency and enabling 3D reconstruction of the generated frames.

**Strengths:**

- The Cross-View Attention and Cross-Frame Attention mechanisms are intuitively designed and effectively enhance temporal consistency. The incorporation of 3D attention particularly strengthens this aspect.

- The joint training strategy is commendable for its use of diverse dataset sources, including monocular videos and multi-view videos, which contributes to the robustness of the model.

**Weaknesses:**

- The video content presented demonstrates low dynamics (object), which may limit the ability to fully showcase the capabilities of the proposed method.

- The contribution of this work appears to be incremental. The proposed method shares significant similarities with CVD, suggesting that the innovations may be more evolutionary than revolutionary.

- The trajectories showcased in the supplementary demo are relatively simple, which raises questions about the model's ability to generalize to more complex scenarios. Further demonstration on a broader range of data would help to substantiate the method's effectiveness.

**Questions:**

- The camera conditions are normalized to a unit scale, can this approach still achieve variable speed camera control similar to MotionCtrl?

- Is it sufficient for the filter method used in Particle-sfm for camera pose estimation to rely on the number of points for accurate pose estimation in videos with significant object motion?

- It would be beneficial to include data samples from various sources to enhance understanding. Additionally, filtering out poor quality samples is necessary.

---

> ### Author Response · Authors · 2024-11-26
>
> We thank the reviewer UoYX for the detailed comments and actionable suggestions. However, we are not able to agree with the reviewer’s over-concerning about our contribution. Thanks to our novel architectural design and our joint training strategy, our proposed framework outperforms all prior or concurrent camera-controlled video generation methods in terms of object motion quality or camera pose accuracy. Detailed responses are provided below.
>
> **Q1. The video content presented demonstrates low dynamics (object), which may limit the ability to fully showcase the capabilities of the proposed method.**
>
> We have included generated videos with large object motion in the supplementary. We have also added some of them to the main draft as suggested (Fig. 11). We invite the reviewer the evaluate these results in a comparative context. As clearly demonstrated in the supplementary videos, baseline methods fail to produce any motion, remaining purely static. In contrast, our method produces vivid object motion while maintaining camera pose control accuracy.
>
> **Q2. The contribution of this work appears to be incremental. The proposed method shares significant similarities with CVD, suggesting that the innovations may be more evolutionary than revolutionary.**
>
> We’d like to remind the reviewer that CVD is a concurrent work, as mentioned by the ICLR 2025 policy. We have already included their results for comparison in our supplementary materials. It can be clearly seen that our method produces significantly more accurate camera control and much clearer reconstructed 3D scenes. Below we provide detailed analysis and comparisons.
>
> - **Task** : CVD is a concurrent work designed for text-to-video generation, while our focus is image-to-video generation. We have compared qualitatively and shown great improvement in camera pose accuracy and 3D reconstruction quality.
> - **Arch**: The CVD framework is limited to leveraging only two views. In contrast, our proposed cross-view attention adopts a fundamentally different approach, allowing us to flexibly train and test with an arbitrary number of views. This capability enables joint training on both monocular and multi-view video datasets and facilitates generalization to four views during testing. By sampling four views simultaneously in a single pass, our approach is more efficient and accurate compared to CVD, which requires enumerating all two-view pairs at each diffusion step.
> - **Data**: We make the first attempt to jointly train on real multi-view and monocular videos, instead of using homography augmented data in CVD.The reliance on pseudo multi-view data without realistic camera motion in CVD makes it challenging to achieve complex camera motions. Furthermore, the black regions introduced by homography augmentation in CVD often manifest as black artifacts in their generated outputs, as shown in our supplementary materials.
>
>
> **Q3. The trajectories showcased in the supplementary demo are relatively simple, which raises questions about the model's ability to generalize to more complex scenarios. Further demonstration on a broader range of data would help to substantiate the method's effectiveness.**
>
> We have included more complex cameras in the supplementary materials in Figure F. We’d like to mention that neither the camera poses nor the testing images are seen during the model training, indicating outstanding generalization ability of the proposed model.
>
> **Q4. The camera conditions are normalized to a unit scale, can this approach still achieve variable speed camera control similar to MotionCtrl?**
>
> The camera speed control in MotionCtrl is implemented by scaling the camera translation along trajectories. However, because our framework normalizes the translation of the entire trajectory, it does not support changing the scale of the trajectory in the exact same way. This limitation is acknowledged in our limitations section.
>
> It is important to note that the scale implementation in MotionCtrl is not metric-scale. Their trajectory scaling relies on cameras estimated by COLMAP, which assigns an arbitrary scene scale for each separate COLMAP run. As a result, MotionCtrl does not achieve valid speed control. Moving forward, we aim to explore the use of metric-scale data to enable more precise camera speed control.
>
> That said, our current framework does support changing the speed of camera motion. By setting the camera position of the final frame to a fixed distant location, we can arbitrarily adjust the speed of the cameras for the remaining frames, achieving variable-speed camera control.

---

> > ### Author Response · Authors · 2024-11-26
> >
> > **Q5. Is it sufficient for the filter method used in Particle-sfm for camera pose estimation to rely on the number of points for accurate pose estimation in videos with significant object motion?**
> >
> > We have provided visualizations of the annotations of Particle-sfm in Fig. 12. We observe that some videos are captured with almost static cameras, making it extremely challenging for SfM workflows to generate point clouds with reasonable shapes. Notably, these videos still contain a relatively large number of points in the point cloud, indicating that the number of points alone is not a perfect metric.
> > Empirically, we found that the camera pose classification module based on optical flow successfully identifies problematic cases, such as the one in Fig. 12. Consequently, our final data mixture is curated through a rigorous filtering pipeline that incorporates aesthetic scores, OCR, and camera motion classification using optical flow.
> >
> >
> > **Q6. It would be beneficial to include data samples from various sources to enhance understanding. Additionally, filtering out poor quality samples is necessary.**
> >
> > Thanks for the suggestion. We have included examples from each source in Fig. 13.

---

> > > ### Comment · Reviewer_UoYX · 2024-11-27
> > >
> > > Thanks for the author's reply, I think the use of this multi-type data set is good, but the overall model design and effect are relatively weak. The final score will refer to the suggestions of other reviewers, and I will keep it now.

---

> > > > ### Author Response · Authors · 2024-11-27
> > > >
> > > > Thank you for acknowleding our contributions in the data pipelines. Regarding the object motion effects, we have included an additional side-by-side comparison of our results with those of SVD, our base model, in Fig. 14. As observed, SVD exhibits `low-dynamics` highlighted by the reviewer. In comparison, our method is able to generate realistic object motion. This suggests that the `low-dynamics` behavior is potentially attributable to the limited capacity and the frame length constraints of the base model. In future works, we plan to explore advanced base models with enhanced object motion generation capabilities.
> > > >
> > > > We fully acknowledge that this field presents numerous unsolved challenges. While our approach does not claim to address all of these challenges, it represents a pioneering method that achieves high-quality, camera-controllable multi-view video generation—an accomplishment, to our knowledge, unmatched by any existing method.

---

> > > > ### Author Response · Authors · 2024-12-02
> > > >
> > > > Dear Reviewer UoYX,
> > > >
> > > > As the author-reviewer discussion deadline is approaching, we wish to emphasize that **the factors you prioritize most, such as model design, object motion, and generalization capabilities, are precisely the areas where Cavia excels**.
> > > >
> > > > - **Model design:** Our novel model design provides flexibility during both training and testing, enabling the use of a joint training strategy. This design ensures **superior consistency** compared to the concurrent work, CVD, which is limited to processing two views at a time. Furthermore, our flexible architecture leverages the Objaverse dataset to support **both geometric consistency and motion synchronization**, capabilities unattainable with CVD's pseudo-multi-view data augmented from single-view videos.
> > > >
> > > > - **Object motion:** Fig. 14 provides comparisons with our base model SVD, demonstrating that Cavia can generate object motion strength comparable to SVD. It is important to note that previous camera-controllable video generation methods based on SVD are limited to producing purely static scenes.
> > > >
> > > >
> > > > - **Generalization ability:** Our supplementary videos contain **generalization to out-of-distribution cameras**, utilizing trajectories from DL3DV-10k, CO3Dv2, and even hand-crafted trajectories. In comparison, MotionCtrl and CameraCtrl demonstrate results exclusively on in-distribution camera trajectories derived from RealEstate10k. Moreover, Cavia leverages Objaverse dataset, along with a carefully designed data mixture and training strategy, to produce results with realistic backgrounds, demonstrating **superior generalization to diverse image content**. In contrast, SV3D, IM-3D, and VideoMV exhibit weaker generalization, producing artificial backgrounds composed solely of uniform colors.
> > > >
> > > > **We kindly request that you consider revisiting your assessment in light of this new evidence.**  Please do not hesitate to contact us if there are other clarifications we can offer. Thanks!

---

> > > > ### Author Response · Authors · 2024-12-03
> > > >
> > > > Dear Reviewer UoYX,
> > > >
> > > > We would like to remind you that the deadline for the author-reviewer discussion is approaching in a few hours. Despite multiple reminders sent previously, we have not yet received a response from you. Could you kindly review our latest reply (if you have not already) to confirm whether it addresses your concerns, or let us know if you have any additional questions? Would you be open to reconsidering your assessment in light of this new evidence?

---

> ### Author Response · Authors · 2024-12-01
>
> Dear Reviewer UoYX,
>
> Thank you for your valuable feedback. To address your concerns about the low object motion in our results, we have provided **a new Fig. 14, which offers a direct comparison between our method and our base model, SVD**. As highlighted by Reviewer T6Cg, **the object motion generated by our approach is comparable to that of SVD**. Furthermore, Cavia introduces critical capabilities, such as precise camera control and motion synchronization for multi-view video generation, which are not supported by SVD. This comparison demonstrates that **the level of object motion in our results reflects the inherent characteristics of SVD**, which we selected as our base model for its status as the best-performing image-to-video model available at the time of our study. Moving forward, we are eager to integrate our proposed innovations into more advanced base models as they become publicly accessible.
>
> While methods like Sora from OpenAI are recognized for producing larger object motion in standard monocular video generation, they are not designed for camera-controllable multi-view video generation. **Our extensive experiments validate that Cavia significantly outperforms existing research works in this domain.**
>
> We greatly appreciate your thoughtful evaluation and hope that the additional qualitative comparisons and the favorable assessments by other reviewers (T6Cg, EfvF) provide a clearer understanding of our contributions. **We kindly request that you consider revisiting your assessment in light of this new evidence.** We have put significant effort into building Cavia and believe our work offers substantial value to the research community, particularly in advancing controllable multi-view video generation.

---

### Official Review · Reviewer_FTwe · 2024-10-30

**Soundness:** 3
**Presentation:** 3
**Contribution:** 2
**Rating:** 5
**Confidence:** 4

**Summary:**

This paper presents a novel method for camera-controllable multi-view video generation. Leveraging a pre-trained I2V model, the method introduces camera-conditioned Plücker coordinates in the latent space for conditional control. Additionally, it extends spatial attention into cross-view attention and promotes 1D temporal attention into cross-frame 3D attention. These enhancements have been evaluated and found beneficial for achieving cross-view and cross-frame consistency, as well as precise camera control.

To address the challenge of limited multi-view video data, an effective joint training strategy is proposed, utilizing a curated mixture of static, monocular dynamic, and multi-view dynamic videos. Experiments demonstrate the superiority of the proposed method over existing competitors in terms of multi-view consistency and camera control precision. However, it is worth noting that most of the demonstrated examples pertain to specific domains, involving minimal dynamics and simple backgrounds, which raises questions about the method's performance in more general scenarios.

**Strengths:**

1. From the qualitative and quantitative results, it is evident that this method indeed offers superior camera trajectory control accuracy. This advantage is attributed to the benefits brought by 3D temporal attention (maybe also fulltuning) and the carefully collected training dataset.

2. To overcome the scarcity of multi-view video dataset, a curated mixture of existing available static, monocular dynamic, and multi-view dynamic videos is exploited via joint training scheme. This provides a solution for addressing data issues in future research on this task.

3. The experimental evaluation and comparison are comprehensive, evaluating the geometric accuracy of camera control from multiple perspectives. It demonstrates noticeable superiority of the proposed method.

**Weaknesses:**

1. The technical novelty is mild. Adopting plucker embedding for camera control is not new. Extending spatial attention for cross-view attention is also straightforward. Anyway, it is inspiring that 3D attention instead of 1D temporal attention is highly desired to help camera control accuracy, and this is well validated by multiple experiments.


2. One of the major weakness is the less convincing results.
The most important aspects of multi-view video generation are motion synchronization and geometric consistency. However, most of the examples presented in the paper involve minimal object movement (either a single object with a simple, weakly moving background or entirely static scenes), and the camera angles do not vary significantly, with most shots taken from the same side of the object. This makes it difficult to evaluate the proposed method's capabilities in this task.

For example, I would like to see examples like this: "on a street with vehicles and pedestrians, a horse is running, with one camera capturing a global view from behind the horse and another camera tracking from the side". Showing more examples like this would be more convincing and demonstrate the value of controllable multi-view video generation.


3. In my understanding, making use of monocular videos could promote the model's generalization. But most of the cases in the paper, are close to the style of Objeverse dataset. Is it because the fulltuning destroyed the capabiliy of SVD in generating realistic style and scene? If so, this should be claimed as a limitation, especially compared to those adapter plugin methods like MotionCtrl and CameraCtrl.


4. From the results in Fig. 8, it appears that while the introduction of monocular video increases generalization ability, such as maintaining the content of the input images, it also affects the accuracy of multi-view camera control. In the second row of (b), the shooting direction for the fish and the fox did not turn as expected.

**Questions:**

1. In Table 3, I'm curious why the impact of "w/o cross-view" is even smaller than that of "w/o cross-frame." Intuitively, cross-view should be an essential component for multi-view setups (without it, it's challenging to maintain multi-view consistency and motion synchronization), whereas cross-frame primarily enhances the accuracy. Does this result imply that most of the outcomes are actually static, and similar results could be achieved through independent monocular camera control?

2. Is your model fully tuned? What is the parameter count?

3. Are the results of the ablation study in Table 3 based on monocular camera control or multi-view camera control? It would be advisable to evaluate these two settings separately.

---

> ### Author Response · Authors · 2024-11-26
>
> We thank the reviewer FTwe for the reviewing time and constructive suggestions. However, we respectfully disagree with the reviewer’s concerns regarding the scale of object motion. We encourage the reviewer to evaluate our results in a comparative context. In the supplementary videos, it is clearly shown that the other camera-controlled video generation frameworks are only able to generate pure static videos, whereas our approach demonstrates vivid and dynamic object motion.
>
> **Q1. most of the examples presented in the paper involve minimal object movement.**
>
> We have examples with large object motion in the supplementary. We invite the reviewers to evaluate our results in a comparative context. While the results presented in the paper do not showcase extremely large object motion, it is evident that baseline methods are restricted to producing purely static outputs, regardless of the content. In contrast, our proposed method addresses this limitation in controllable multi-view video generation by presenting vivid and dynamic object motions.
>
> **Q2. For example, I would like to see examples like this: "on a street with vehicles and pedestrians, a horse is running, with one camera capturing a global view from behind the horse and another camera tracking from the side". Showing more examples like this would be more convincing and demonstrate the value of controllable multi-view video generation.**
>
> We agree that the proposed prompt is more encouragingadvancing the field of controllable multi-view video generation. However, it exceeds the capabilities of our current base model, Stable Video Diffusion (SVD). When working on this project, no superior open-source image-to-video models were available. Looking ahead, we are eager to integrate our proposed innovations into more advanced base video models as they become publicly accessible.
>
> **Q3. Most of the cases in the paper, are close to the style of Objeverse dataset. Is it because the fulltuning destroyed the capabiliy of SVD in generating realistic style and scene? If so, this should be claimed as a limitation, especially compared to those adapter plugin methods like MotionCtrl and CameraCtrl.**
>
> We have provided results with complex scenes (e.g. street views, natural scenic and indoor scenes) in the supplementary. It is important to note that our evaluation methodology fundamentally differs from prior works that focus exclusively on the Objaverse dataset. Unlike these approaches, all of our evaluations involve rich backgrounds, requiring the model to accurately comprehend the geometry of the scenes.
>
> Regarding the MotionCtrl and CameraCtrl approaches, the image-to-video models released by their authors generate only **purely static** outputs. This limitation is undesirable for controllable multi-view video generation, where object motion plays a critical role. In comparison, our method demonstrates the ability to generate vivid object motion, guided by precise camera instructions, even in complex scenes.
>
>
> **Q4. From the results in Fig. 8, it appears that while the introduction of monocular video increases generalization ability, such as maintaining the content of the input images, it also affects the accuracy of multi-view camera control. In the second row of (b), the shooting direction for the fish and the fox did not turn as expected.**
>
> Thanks for raising this good question. We’d like to clarify that the differences in Fig. 8 (a) and (b) are not related to the accuracy of multi-view camera control.
>
> - For the fox example, the variation in the fox’s movement across different sampled videos arises because our framework does not control object motion. However, examining the viewpoint change around the fox’s feet reveals that the camera motion remains consistent.
>
> - For the fish example, the behavior described as "did not turn as expected" occurs because our camera control is not metric-scale. Instead, the scene’s scale is implicitly determined by the model. It is clearer shown in the dynamic video that the Fish(b) exhibits a smaller translation scale compared to Fish(a). Therefore, the coral reef appears to rotate differently.
>
> Due to the lack of a high-quality open-source metric-scale depth model as well as high-quality object motion annotations, we leave such explorations for future work. Nonetheless, as shown in Tab. 3, the introduction of monocular video does not negatively impact the multi-view camera control accuracy.

---

> > ### Author Response · Authors · 2024-11-26
> >
> > **Q5. In Table 3, why the impact of "w/o cross-view" is even smaller than that of "w/o cross-frame." Does this result imply that most of the outcomes are actually static, and similar results could be achieved through independent monocular camera control?**
> >
> > Thanks for bringing up this issue. The model variant "w/o cross-frame" also excludes the ``cross-view attention’’. We have clarified this in our revised draft in Tab. 4. Removing the `cross-frame attention’ will severely deteriorate the visual quality of the generated frames, particularly when the camera viewpoints deviate substantially from the origin. This results in dramatically worse performance metrics.
> >
> >
> >
> > **Q6. Is your model fully tuned? What is the parameter count?**
> >
> > Our model is fully finetuned, the parameter count is roughly the same as SVD, with around 1.5B parameters.
> >
> > **Q7. Are the results of the ablation study in Table 3 based on monocular camera control or multi-view camera control? It would be advisable to evaluate these two settings separately.**
> >
> > The camera pose accuracy are evaluated under 2-view camera control setting. The FID and FVD metrics can be considered to be in monocular setting, because they measure the perceptual quality of the generated frames with respect to the ground truth frames. Consequently, these results for model variants without the cross-view attention (e.g. `w/o Plucker`, `w/o Cross-frame`, `w/o Cross-view`) also indicate the quality of monocular camera control.

---

> > ### Comment · Reviewer_FTwe · 2024-11-26
> >
> > Thank you for your responses.
> >
> > After reviewing the point-to-point responses, my main concern regarding the results still remains. While the proposed method demonstrates impressive accuracy in camera control, its performance as a multi-view video generation model is not entirely convincing. Specifically, the current results do not adequately reflect the proposed method's capability as a solid multi-view video generation technique. So, I will keep my rating.

---

> > > ### Author Response · Authors · 2024-11-26
> > >
> > > We appreciate your prompt responses and acknowledgment of our expertise in camera control.
> > >
> > > Regarding 'multi-view video generation,' we would like to offer a gentle clarification. As you mentioned above, the critical challenge in multi-view video generation lies in ensuring synchronized object motion across different viewpoints. To the best of our knowledge, no prior or concurrent work achieves comparable performance to ours in this domain. This is particularly true for the even more demanding task of camera-controllable multi-view video generation.
> > >
> > > We fully agree that this field presents numerous unsolved challenges. While our approach does not claim to resolve all these challenges, it represents a first-of-its-kind method, achieving high-quality camera-controllable multi-view video generation that, to our knowledge, no other method has accomplished.

---

> > > ### Author Response · Authors · 2024-11-29
> > >
> > > Dear Reviewer FTwe,
> > >
> > > We have included side-by-side comparisons of Cavia and our base model, Stable Video Diffusion (SVD) in Fig. 14. The frames generated by our method exhibit rich object motion that is comparable to, if not better than, that of SVD. For example, our penguin, giraffe, and bird examples are behaving realistically, while SVD presents limited object motion.
> > >
> > > Will these additional results address your concerns regarding object motion, given that Cavia demonstrates comparable object motion strength to SVD? Our extensive experiments show that Cavia outperforms all other methods in multi-view video generation. Could you please clarify any other concerns or issues that lead you to believe Cavia may not be a solid solution for multi-view video generation?

---

> > > ### Author Response · Authors · 2024-12-01
> > >
> > > Dear Reviewer FTwe,
> > >
> > > Thank you for sharing your concerns regarding the performance of Cavia. To address these points, we have provided a **side-by-side comparison with SVD, demonstrating that our model achieves a similar level of object motion strength as its base model**. Additionally, while we acknowledge the impressive object motion generation capabilities of Sora from OpenAI, it is important to note that Sora is not designed for camera-controllable multi-view video generation. In this domain, Cavia represents a significant advancement, as **no existing research achieves comparable performance in camera-controllable video generation**. As shown in our experiments, **previous works are only able to generate pure static scenes**.
> > >
> > > We kindly ask if you would consider revisiting your assessment in light of the new qualitative comparisons and the positive evaluations provided by other reviewers (T6Cg, EfvF). The authors have dedicated substantial effort to developing Cavia and strongly believe in its potential to contribute meaningfully to the research community.

---

> > > > ### Comment · Reviewer_FTwe · 2024-12-02
> > > >
> > > > Thank you for your reply! I fully agree on the importance and value of exploring the dynamic multi-camera video generation task. However, the overall quality of this paper is currently not very strong. With a relatively straightforward or mild technical design, the method demonstrates limited video dynamics and poor generalization capabilities (I suspect this is mainly due to the characteristics of the **objverse** dataset). This means that the exploration of this task has not yet reached an inspiring milestone. If the dataset (including dynamic range and domain diversity) were more comprehensive, whether this approach could achieve promising results remains unknown; at least, it is not evident from the current results.
> > > >
> > > > Based on my experience, although SVD does not have a large overall dynamic range, it is clearly higher than the examples shown in this paper. The authors could select a few examples where SVD achieves a larger dynamic range for comparison.
> > > >
> > > > Therefore, overall, I am inclined to be negative about this paper.

---

> > > > > ### Author Response · Authors · 2024-12-03
> > > > >
> > > > > Dear Reviewer FTwe,
> > > > >
> > > > > We would like to remind you that the deadline for the author-reviewer discussion is just a few hours away. Despite our best efforts to reach out, we have not yet received a response from you. Could you kindly review our latest reply (if you have not already) to confirm whether it addresses your concerns, or let us know if you have any additional questions? Additionally, would you be open to reconsidering your assessment in light of the new evidence presented?

---

> ### Author Response · Authors · 2024-12-02
>
> Thank you for sharing your thoughts and highlighting the importance of our multi-view video generation task. We appreciate your feedback and would like to address your concerns to clarify any misunderstandings:
>
> - **Generalization ability:**
>     - We would like to emphasize that MotionCtrl and CameraCtrl demonstrate results exclusively on in-distribution camera trajectories derived from RealEstate10k. In contrast, we additionally present **out-of-distribution results**, utilizing trajectories from DL3DV-10k, CO3Dv2, monocular videos, and even hand-crafted trajectories. This represents a **significant improvement in generalization to diverse camera trajectories** compared to prior work.
>     - The Objaverse dataset serves as a critical data source for supporting our multi-view video motion generation capabilities. Our model leverages this dataset, along with a carefully designed data mixture and training strategy, to produce results with **realistic backgrounds**, demonstrating **superior generalization to diverse real image content**. In contrast, SV3D, IM-3D, and VideoMV exhibit poorer generalization, generating artificial backgrounds composed solely of uniform colors.
> - **Data source:**
>     - **Previous camera-controllable methods are limited to generating static results, a challenge further compounded in multi-view scenarios due to the absence of high-quality multi-view video datasets.** A concurrent work, CVD, attempts to address this limitation by augmenting pseudo-multi-view data from single-view videos. However, as shown in our supplementary materials, this approach exhibits poorer consistency compared to our method.
>     - **We propose leveraging Objaverse objects to ensure precise camera controllability and synchronized object motion across multi-view videos**. Unlike previous methods such as SV4D, 4DGen, and DreamGaussian4D, which are limited to producing foreground elements, our approach demonstrates exceptional **geometric consistency for both foreground and background**. Furthermore, our results **maintain synchronization when object motion is present**.
> - **Object motion strength:**
>     - While SVD allows a wide range of motion strengths as a hyperparameter, **we fixed the motion strength to 127** (which is the recommended value) across all experiments, including the fine-tuning of Cavia. This ensures that Fig. 14 provides a fair comparison of motion strength capabilities before and after fine-tuning, highlighting Cavia as a robust and effective multi-view video generation framework. We also emphasize that **previous camera-controllable video generators based on SVD were limited to producing static results, underscoring Cavia’s contribution to the field**.
>
> We hope this additional evidence clarifies your concerns. To the best of our knowledge, no other research work achieves performance comparable to Cavia, underscoring its significant contribution to the research community. We appreciate your high expectations and rigorous standards for Cavia. However, we wish to emphasize that **the factors you prioritize most, such as object motion and generalization capabilities, are precisely the areas where Cavia excels.** Our approach outperforms all related works by a substantial margin in the areas you mentioned.

---

### Official Review · Reviewer_EfvF · 2024-10-31

**Soundness:** 4
**Presentation:** 3
**Contribution:** 3
**Rating:** 8
**Confidence:** 4

**Summary:**

This paper introduces CAVIA, a framework that advances video generator by adding camera controllability and enabling 4D generation. Specially, it focuses on generating multi-view consistent videos, allowing precise control over camera movement while maintaining object and scene consistency across frames. The Plücker embedding is adopted to inject camera features to the video generator. The cross-frame attention is used to enhancing temporal consistency within each video clip, and the cross view attention is utilized to enhance the content consistency between across different video clips.
CAVIA is also novel in its usage of multiple data sources, including static and dynamic, multi-view datas. This methods tested on various datasets and compared with some existing methods, demonstrating its state-of-the-art performance.

**Strengths:**

1. Writing is good, and is easy to follow.
2. The model is simple and effective, and easy to use the existing pre-trained video generation models.
3. The data preparation process is logical and effective, it leverages a mix of statics multi-view, monocular videos, and synthetic multi-view datas. The model benefits from such a broader range of training data, enhancing its generalization and robustness.
4. Extensive experiments demonstrate the effectiveness of each design choice.

**Weaknesses:**

The main weaknesses are in the experiment part:
1. What are the "general" videos come from? Compared to the RealEstate10K dataset, does the "general" videos have larger camera motion? And does the dynamic complexity is limited?
2. In Table 2, there should be some quantitative comparisons with the CVD model.
3. In the provided videos, the object dynamic degree is limited, can you provide a comparison on the object dynamic degree between CAVIA and the base video generation model SVD?

**Questions:**

See weaknesses

---

> ### Author Response · Authors · 2024-11-26
>
> We thank the reviewer EfvF for the favorable assessments and detailed comments. Our responses to your concerns are listed as follows.
>
> **Q1. What are the "general" videos come from? Compared to the RealEstate10K dataset, does the "general" videos have larger camera motion? And does the dynamic complexity is limited?**
>
> The ``general’’ videos used for evaluation are sourced from our monocular video dataset. Based on our observations, these videos typically exhibit smaller camera motion compared to the RealEstate10K dataset. This difference arises because most YouTube videos are not characterized by simultaneous large object motion and extensive camera motion. However, these videos contain rich object motion, in contrast to RealEstate10K, where the majority of scenes are static.
>
> **Q2. In Table 2, there should be some quantitative comparisons with the CVD model.**
>
> Unfortunately, the CVD model has not been publicly released, limiting our comparisons to qualitative analysis. We have provided the comparisons in our supplementary materials (Fig. C and Fig. D). It can be clearly seen that our generated videos enjoy better consistency and improved pose accuracy. Moreover, our generated videos can be reconstructed into clearer and more coherent 3D scenes.
>
> **Q3. In the provided videos, the object dynamic degree is limited, can you provide a comparison on the object dynamic degree between CAVIA and the base video generation model SVD?**
>
> Thanks for raising this good question. We also observe that two view generation reduces the object motion to some extent. This can be seen in our ablation study (“Ablation on cross-view attention”). Specifically, we find that our model variant without the `cross-view attention` exhibits larger object motion compared to the one with `cross-view attention`.
>
> We hypothesis that this behavior may be attributed to the model’s capacity. Since the `cross-view attention` requires our model to generate synchronized object motions, it tends to produce only those motions that can be confidently synthesized from multiple viewpoints. This task is inherently more challenging than monocular camera-controllable video generation and even more so than arbitrary video generation without camera pose, as seen in Stable Video Diffusion (SVD).
>
> We have also provided a side-by-side comparison between Cavia and SVD in Fig. 14. The frames generated by our method exhibit rich object motion that is comparable to, if not better than, that of SVD. For example, our penguin, giraffe, and bird examples are behaving realistically, while SVD presents limited object motion.

---

> > ### Comment · Reviewer_EfvF · 2024-11-30
> >
> > I thank the author's time for providing extra qualitative comparisons. I think Cavia's exploration of applying a variety of different types of data to train a camera-controllable multi-view video diffusion model is meaningful and insightful. I still argue for its acceptance.

---

> > > ### Author Response · Authors · 2024-11-30
> > >
> > > We greatly appreciate reviewer EfvF’s positive assessment and recognition of our contribution to the camera-controllable multi-view video generation community. Developing Cavia required substantial effort, and we are confident in its significance. Your encouraging feedback is highly motivating for the authors.

---

> ### Author Response · Authors · 2024-11-27
>
> Dear Reviewer EfvF,
>
> As the author-reviewer discussion deadline is approaching, could you please take a look at the authors' rebuttal (if not yet) and see if it addressed your concerns or if you have any further questions?
>
> Best,
>
> Authors

---

### Official Review · Reviewer_T6Cg · 2024-11-07

**Soundness:** 2
**Presentation:** 1
**Contribution:** 2
**Rating:** 5
**Confidence:** 3

**Summary:**

This paper presents a novel framework, Cavia, designed for generating multi-view videos with precise camera control. The authors introduce view-integrated attention mechanisms that enhance 3D consistency across perspectives. Additionally, Cavia is trained on a curated blend of static, monocular dynamic, and multi-view dynamic videos. Cavia demonstrates superior performance over baseline methods in both monocular video generation and cross-view consistency.

**Strengths:**

1. Novel Framework for Camera-Controllable Video Generation: This paper introduces an innovative framework aimed at generating videos with precise camera control.

2. Superior Performance over Baselines: The proposed method demonstrates an  improvement over baseline models across key performance metrics.

3. Good Monocular Setting Results: Under a monocular setting, the experimental results exhibit substantial improvements in the generated videos.

**Weaknesses:**

1. Limited Distinction in Visualization Results (Figure 5): The visual examples provided in Figure 5 lack noticeable differences, which undermines the claimed improvements. This inconsistency between visual and metric-based results calls into question the interpretability and significance of the enhancements.

2. Poor Presentation: (i) The method section is minimal, with only three formulars, all of which are borrowed from references. The lack of detailed descriptions of the pipeline leaves the approach unclear and less accessible for readers. (ii) The paper includes non-standardized expressions, such as "(B V F C H W)" in Line 222. Additionally, Section 4 is ambiguously presented, making it difficult to comprehend.

**Questions:**

1. Visualization vs. Metric-Based Improvement: In Figure 5, the differences among the results appear minimal. What factors contribute to the significant metric-based improvements under the 2-view setting despite the subtle visual changes?

2. GPU Memory Usage of Cross-Frame Attention: Given the extended token length in the cross-frame attention module, does this component impose a significant demand on GPU memory?

---

> ### Author Response · Authors · 2024-11-26
>
> We thank the reviewer T6Cg for their time and actionable suggestions. We have updated the presentation of sections 3 and 4 as suggested. Regarding the concern about our method being minimal, we would like to emphasize that our contributions include both advancements in model architecture and the curation and analysis of data. For other questions, we also provide detailed responses as below.
>
> **Q1. The visual examples provided in Figure 5 lack noticeable differences, which undermines the claimed improvements. What factors contribute to the significant metric-based improvements under the 2-view setting despite the subtle visual changes?**
>
> We include examples with more noticeable differences in the supplementary and have incorporated them into the updated draft (Fig. 11). We have also included side-by-side comparisons between our method and the base model (SVD) in Fig. 14 to validate the object motion strength we generate. Our method outperforms the camera-controllable baseline methods by a large margin in two aspects:
>
> - **Object motion**:  When examining dynamic videos rather than individual static frames, it is evident that the baseline methods generate predominantly static results. In contrast, our method demonstrates vivid and realistic object motion.
>
> - **Pose accuracy**: In camera-controllable video generation, achieving precise control over camera pose is critical. While humans are adept at perceiving the general direction of camera motion, they often struggle to assess its accuracy. To address this, we systematically evaluate the pose accuracy of the generated frames using COLMAP and SuperGlue. Our results show that our method achieves the most accurate camera pose control among all compared approaches.
>
> **Q2. Poor Presentation: (i) The method section is minimal. (ii) The paper includes non-standardized expressions, such as "(B V F C H W)" in Line 222.  (iii) Section 4 is ambiguously presented, making it difficult to comprehend.**
>
> We have updated our sections 3 and 4 as suggested by the reviewer. Please kindly note that our proposed framework involves innovations both in terms of the model architecture and the data construction. We provide a detailed analysis of the joint training strategy that drives our model's performance, as well as the data curation workflow that supports this strategy.
>
> **Q3. Given the extended token length in the cross-frame attention module, does this component impose a significant demand on GPU memory?**
>
> Our proposed Cross-frame Attention requires extra GPU memory compared to the vanilla 1D temporal attention used in Stable Video Diffusion. However, it’s not a significant demand. During training, our model comfortably fits into an A100 GPU with 80G memory.

---

> > ### Comment · Reviewer_T6Cg · 2024-11-28
> >
> > Thanks for the response.
> > I did not see much improvement on the method illustration.
> > What does each row represent in fig 11? No explanation on the figure.
> > I did not see the claimed "outperforms the baseline methods by a large margin" at least in figure 14.

---

> > > ### Author Response · Authors · 2024-12-02
> > >
> > > Dear Reviewer T6Cg,
> > >
> > > As the author-reviewer discussion deadline approaches, we wish to emphasize that the factors you prioritize most, such as **object motion**, are precisely the areas in which Cavia excels.
> > >
> > > Fig. 14 provides comparisons with our base model SVD, demonstrating that **Cavia can generate object motion strength comparable to SVD**. **Notably, previous camera-controllable video generation methods based on SVD (e.g., MotionCtrl and CameraCtrl) are constrained to producing purely static scenes**.
> > >
> > > Our significant metric-based improvements stem from the exceptional generalization ability of our framework. The supplementary videos showcase **generalization to out-of-distribution cameras** by employing trajectories from DL3DV-10k, CO3Dv2, and even hand-crafted trajectories. In comparison, MotionCtrl and CameraCtrl demonstrate results exclusively on in-distribution camera trajectories derived from RealEstate10k. Additionally, Cavia leverages Objaverse dataset, combined with a carefully designed data mixture and training strategy, to produce results with realistic backgrounds, demonstrating **superior generalization to diverse image content**. In contrast, SV3D, IM-3D, and VideoMV also train on Objaverse dataset, but exhibit weaker generalization, producing artificial backgrounds composed solely of uniform colors.
> > >
> > > **Would you be open to reconsidering your assessment in light of this new evidence? To the best of our knowledge, no other research work matches Cavia's performance, underscoring its significant contribution to the research community.** Please do not hesitate to contact us if further clarifications or additional information are required. Thank you!

---

> > > ### Author Response · Authors · 2024-12-03
> > >
> > > Dear Reviewer T6Cg,
> > >
> > > We would like to remind you that the deadline for the author-reviewer discussion is just a few hours away. Despite our best efforts to follow up, we have not yet received your response. Could you kindly review our latest reply (if you have not already) to confirm whether it addresses your concerns or let us know if you have any additional questions? Furthermore, we would greatly appreciate it if you could reconsider your assessment in light of the new evidence we have presented.

---

> ### Author Response · Authors · 2024-11-27
>
> Dear Reviewer T6Cg,
>
> As the author-reviewer discussion deadline is approaching, could you please take a look at the authors' rebuttal (if not yet) and see if it addressed your concerns or if you have any further questions?
>
> Best,
>
> Authors

---

> ### Author Response · Authors · 2024-11-28
>
> Thanks for the prompt response. Can you please clarify which part is confusing about the method illustration? We will be happy to revise the draft accordingly.
>
> In Fig. 11, we provide 2-view videos generated by Cavia. For each set of two rows, the first row and the second row refer to the two different views.
>
> In Fig. 14, we compare the motion strength of our method against SVD, which serves as our base model. While our approach performs on par with SVD in terms of motion strength, it additionally enables precise camera control and supports motion synchronization for multi-view video generation. In contrast, SVD is specifically designed for generating standard monocular videos without any control mechanisms. We appreciate your recognition that our method achieves motion strength comparable to our base model.
>
> By saying `Cavia outperforms the baseline methods by a large margin`, we refer to the other camera-controllable video generation methods. Previous camera-controllable video generation methods are all limited to imprecise camera controls and pure static scenes.
>
> Since the deadline for updating the PDF has passed, we have documented these points and will upload a revised version in the future.

---

### Author Response · Authors · 2024-11-26

We sincerely thank all reviewers for their time and detailed comments. **We are pleased that our clear presentation (EfvF), performance (EfvF, UoYX, FTwe), and novelty (T6Cg) have been recognized.** We also noticed some of the reviewers have expressed concerns regarding our experimental results and contributions. Hereby, we would like to reiterate the significance of our work.

Our technical contributions include innovations both in model architecture and in data construction. Specifically, we propose a novel cross-view attention mechanism tailored for the challenging task of camera-controllable multi-view video generation. To support model training, we introduce a data curation workflow and a curated data mixture comprising static 3D scenes, synthetic 3D objects, dynamic 3D objects, and monocular videos.

We demonstrate the superiority of our approach over prior works and the concurrent work CVD in terms of camera pose accuracy, frame perceptual quality, object motion realism, and 3D consistency. Our framework has been rigorously evaluated across diverse test cases. While we acknowledge the object motion generated by our framework is not extremely large, which is likely due to the restricted frame length and the capacity of the base model (Stable Video Diffusion), we encourage reviewers to consider our results in comparison with other methods. Our generated object motion significantly outperforms existing camera controllable image to video generation frameworks, which, as shown in our supplementary videos, are typically restricted to static scenes. In future works, we plan to explore advanced base models with enhanced object motion generation capabilities.

According to reviewers’ suggestions, we have made these several changes (highlighted in blue) to our PDF, summarized below :

- We have updated Sections 3 and 4 to clarify the notations.
- We have added examples with large motion in Fig 11.
- We have clarified our ablation study settings in Tab 4.
- We have added examples of outputs from Particle-SfM in Fig 12.
- We have added visualizations of data samples in Fig. 13.

We would appreciate it if all reviewers could please take a look and finalize their assessments of our work, hopefully more positively. We trust the reviewer and AC discussion will result in an informed and fair decision. We sincerely thank everyone for their valuable time and efforts.

Best,

Paper 2215 Authors

---

### Author Response · Authors · 2024-11-27
**Comparison with the base model SVD regarding object motion**

Dear reviewers,

We have included a side-by-side comparison against our base model (Stable Video Diffusion) in Fig. 14. The frames generated by our method exhibit rich object motion that is comparable to, if not better than, that of SVD. For example, our penguin, giraffe, and bird examples are behaving realistically, while SVD presents limited object motion.

This suggests that the low-dynamics behavior is potentially attributable to the limited capacity and the frame length constraints of the base model. At the time of working on this project, SVD was the best open-source image-to-video model available. In future works, we plan to explore advanced base models with enhanced object motion generation capabilities.

We hope our performance can be evaluated in a comparative context. As shown in our experiments, our method outperforms all prior and concurrent works by a large margin in terms of camera pose accuracy, perceptual quality, and object motion realism. Please let us know if it addresses your concerns or if you have any further questions.

Best,

Authors

---

### Author Response · Authors · 2024-12-02
**Summary of Rebuttal**

Dear Reviewers,

Thank you for your insightful comments that guided us in revising our draft. We are summarizing the major concerns about the paper for quick clarification.

- **[Concerns on object motion.]** Fig. 14 compares Cavia with our base model SVD, showing that Cavia achieves comparable object motion strength. **Notably, previous camera-controllable video generation methods based on SVD (e.g., MotionCtrl and CameraCtrl) are constrained to producing purely static scenes.**
- **[Concerns on generalization ability.]** Cavia enjoys improved generalization ability in terms of camera trajectoris and image content. Our supplementary videos contain **generalization to out-of-distribution cameras**, utilizing trajectories from DL3DV-10k, CO3Dv2, and even hand-crafted trajectories. In comparison, MotionCtrl and CameraCtrl demonstrate results exclusively on in-distribution camera trajectories derived from RealEstate10k. Moreover, Cavia leverages Objaverse dataset, along with a carefully designed data mixture and training strategy, to produce results with realistic backgrounds, demonstrating **superior generalization to diverse image content**. In contrast, SV3D, IM-3D, and VideoMV show limited generalization, producing artificial backgrounds composed solely of uniform colors.

- **[Concerns on the model design compared to the concurrent work CVD.]** Our novel model design enables flexible training and testing with an arbitrary number of views, supporting joint training on both monocular and multi-view video datasets and facilitating generalization to four views during testing. We have provided a qualitative comparison with CVD and shown that Cavia produces clearer results with improved consistency. Unlike CVD, which relies on homography-augmented pseudo multi-view data, **Cavia is the first to jointly train on real multi-view and monocular videos**. This reliance on pseudo multi-view data in CVD, without realistic camera motion, limits its ability to handle complex camera motions and often results in unpleasant black region artifacts caused by augmentation."


We have addressed all the reviewers' questions and thoroughly revised our draft in line with their suggestions. Additionally, we will include further qualitative comparisons in the revised version as recommended. **We kindly urge the reviewers to reassess their ratings in light of the comparative context, considering that no other work has achieved performance on par with Cavia in camera-controllable multi-view video generation.**

**Please do not hesitate to contact us if there are other clarifications we can provide.** Thanks!

Best,

Authors of Paper 2215

---

> ### Author Response · Authors · 2024-12-04
>
> Dear reviewers,
>
> Thank you again for the valuable feedback that significantly enhanced our submission. As the author-reviewer discussion deadline is approaching, we wish to emphasize that **Cavia excels in the very aspects you prioritize most (e.g. model design, object motion, and generalization capabilities) when compared with related research works**. The development of Cavia demanded substantial effort, and we firmly believe it represents a significant contribution to the research community. Notably, no other work has achieved comparable performance in camera-controllable multi-view video generation. Therefore, we respectfully request that you revisit your assessment and evaluate our work fairly, within a comparative framework aligned with the standards of the research community.
>
> Thank you,
>
> Authors of Paper 2215

---

### Meta-Review · Area_Chair_SPn9 · 2024-12-21

**Metareview:**

The paper receives mixed ratings from four reviewers. Most of the reviewers are leaning to reject the submission. They mainly have several concerns regarding this submission, including the insufficient novelty of embedding camera poses as a condition, the unconvincing results shown in the experimental part, and the poor presentation quality. Based on these critical comments, AC recommends a rejection for this time.

**Additional Comments On Reviewer Discussion:**

The reviewers request clarification of the technical novelty, the experimental details, and the results. They are not fully satisfied with the rebuttal of the authors.

---

### Decision · Program_Chairs · 2025-01-22

Reject